# PART-BASED MODELS IMPROVE ADVERSARIAL ROBUSTNESS

**Chawin Sitawarin**[1]        **Kornrapat Pongmala**[1]        **Yizheng Chen**[1]        **Nicholas Carlini**[2]

**David Wagner**[1]

[1] EECS Department, University of California, Berkeley        [2] Google

## ABSTRACT

We show that combining human prior knowledge with end-to-end learning can improve the robustness of deep neural networks by introducing a *part-based* model for object classification. We believe that the richer form of annotation helps guide neural networks to learn more robust features without requiring more samples or larger models. Our model combines a part segmentation model with a tiny classifier and is trained end-to-end to simultaneously segment objects into parts and then classify the segmented object. Empirically, our part-based models achieve both higher accuracy and higher adversarial robustness than a ResNet-50 baseline on all three datasets. For instance, the clean accuracy of our part models is up to 15 percentage points higher than the baseline's, given the same level of robustness. Our experiments indicate that these models also reduce texture bias and yield better robustness against common corruptions and spurious correlations. The code is publicly available at `https://github.com/chawins/adv-part-model`.

## 1 INTRODUCTION

As machine learning models are increasingly deployed in security or safety-critical settings, robustness becomes an essential property. Adversarial training (Madry et al., 2018) is the state-of-the-art method for improving the adversarial robustness of deep neural networks. Recent work has made substantial progress in robustness by scaling adversarial training to very large datasets. For instance, some defenses rely on aggressive data augmentation (Rebuffi et al., 2021) while others utilize a large quantity of extra data (Carmon et al., 2019) or even larger models (Gowal et al., 2021a). These works fall in line with a recent trend of deep learning on "scaling up," i.e., training large models on massive datasets (Kaplan et al., 2020). Unfortunately, progress has begun to stagnate here as we have reached a point of diminishing returns: for example, Gowal et al. (2021a) show that an exponential increase in model size and training samples will only yield a linear increase in robustness.

Our work presents a novel alternative to improve adversarial training: we propose to utilize *additional supervision* that allows for *a richer learning signal*. We hypothesize that an auxiliary human-aligned learning signal will guide the model to learn more robust and more generalized features.

To demonstrate this idea, we propose to classify images with a *part-based model* that makes predictions by recognizing the parts of the object in a bottom-up manner. We make use of images that are annotated with part segmentation masks. We propose a simple two-stage model that combines *a segmentation model* with *a classifier*. An image is first fed into the segmenter which outputs a pixel-wise segmentation of the object parts in a given input; this mask is then passed to a tiny classifier which predicts the class label based solely on this segmentation mask. The entire part-based model is trained end-to-end with a combination of segmentation and classification losses. Fig. 1 illustrates our model. The idea is that this approach may guide the model to attend more to global shape than to local fine-grained texture, hopefully yielding better robustness. We then combine this part-based architecture with adversarial training to encourage it to be robust against adversarial examples.

We show that our model achieves strong levels of robustness on three realistic datasets: Part-ImageNet (He et al., 2021), Cityscapes (Meletis et al., 2020), and PASCAL-Part (Chen et al., 2014). Our part-based models outperform the ResNet-50 baselines on both clean and adversarial accuracy simultaneously. **For any given value of clean accuracy, our part models achieve more than 10**

Figure 1: Our part-based model consists of (1) the part segmenter and (2) a tiny classifier. We train it for the object classification task end-to-end using part-level segmentation labels to improve its robustness.

Figure 2: Accuracy-robustness trade-off of our part model and the ResNet-50 baseline on the Part-ImageNet dataset.

**percentage points higher adversarial accuracy compared to the baseline on Part-ImageNet** (see Fig. 2). This improvement can be up to 25 percentage points in the other datasets we evaluate on (see Fig. 4). **Alternatively, given the same level of adversarial robustness, our part models outperform the baseline by up to 15 percentage points on clean accuracy** (see Table 1).

Our part-based models also improve non-adversarial robustness, without any specialized training or data augmentation. Compared to a ResNet-50 baseline, our part models are more robust to synthetic corruptions (Hendrycks & Dietterich, 2019) as well as less biased toward non-robust "texture features" (Geirhos et al., 2019). Additionally, since our part models can distinguish between the background and the foreground of an image, they are less vulnerable to distribution shifts in the background (Xiao et al., 2021). These three robustness properties are all highly desirable and enabled by the part-level supervision. We believe that our part-based model is the first promising example of how a richer supervised training signal can substantially improve the robustness of neural networks.

## 2 RELATED WORK

### 2.1 ADVERSARIAL ROBUSTNESS

Adversarial training (Madry et al., 2018) has become a standard method for training robust neural networks against adversarial examples. Many improvements on this technique have been proposed (Zhang et al., 2019; Xie et al., 2019; Pang et al., 2019; Huang et al., 2020; Qin et al., 2019; Rice et al., 2020; Wong et al., 2020; Hendrycks et al., 2019; Kireev et al., 2021). Among these, TRADES (Zhang et al., 2019) improves the trade-off between robustness and clean accuracy of adversarial training. More recent state-of-the-art methods focus on improving the adversarial robustness through scales. Carmon et al. (2019) and Gowal et al. (2021a) rely on a large number of unlabeled training data while others utilize large generative models for data augmentation (Rebuffi et al., 2021) or synthetically generating more training samples (Gowal et al., 2021b; Sehwag et al., 2022).

These works follow a recent trend of "large-scale learning from weak signals," which stemmed from recent progress on vision language models such as CLIP (Radford et al., 2021). The improvement from scaling up, however, has started to reach its limit (Gowal et al., 2021a). We take a different route to improve robustness. Our part-based models utilize supervision and high-quality part segmentation annotations to improve robustness without using more training samples or complex data augmentation.

### 2.2 PART-BASED MODELS

Part models generally refer to hierarchical models that recognize objects from their parts in a bottom-up manner, e.g., Deformable Part Models (Endres et al., 2013; Felzenszwalb et al., 2010; Chen et al., 2014; Girshick et al., 2015; Cho et al., 2015). Historically, they are most often used in human recognition (Chen & Yuille, 2014; Gkioxari et al., 2015; Xia et al., 2017; Ruan et al., 2019) and have shown success in fine-grained classification (Zhang et al., 2018; Bai et al., 2019) as well as pose estimation (Lorenz et al., 2019; Georgakis et al., 2019). We revisit part-based models from the robustness perspective and design a general model that can be trained end-to-end without any feature engineering. Our technique is also agnostic to a particular type of object.

Several works have explored part-based models in the context of adversarial robustness. Freitas et al. (2020) detect adversarial examples by using a Mask R-CNN to extract object parts and verify that

they align with the predicted label. If the predicted and the expected parts do not align, the input is regarded as an attack. However, unlike our work, their scheme is not evaluated against an adaptive adversary. Chandrasekaran et al. (2019) propose a robust classifier with a hierarchical structure where each node separates inputs into multiple groups, each sharing a certain feature (e.g., object shape). Unlike our part model, its structure depends heavily on the objects being classified and does not utilize part segmentation or richer supervision. A concurrent work by Li et al. (2023) also investigates segmentation-based part-based models for robustness.

# 3 PART-BASED MODELS

## 3.1 GENERAL DESIGN

**Data samples.** Each sample $(x, y)$ contains an image $x \in \mathbb{R}^{3 \times H \times W}$ and a class label $y \in \mathcal{Y}$, where $H$ and $W$ are the image's height and width. The *training dataset* for part models are also accompanied by segmentation masks $M \in \{0, 1\}^{(K+1) \times H \times W}$, corresponding to $K + 1$ binary masks for the $K$ object parts $(M_1, \ldots, M_k)$ and one for the background $(M_0)$.

**Architecture.** Our part-based model has two stages: the segmenter $f_{\text{seg}} : \mathbb{R}^{3 \times H \times W} \to \mathbb{R}^{(K+1) \times H \times W}$ and a tiny classifier $f_{\text{cls}} : \mathbb{R}^{(K+1) \times H \times W} \to \mathbb{R}^C$. The overall model is denoted by $f := f_{\text{cls}} \circ f_{\text{seg}}$. More specifically, the segmenter takes the original image $x$ as the input and outputs logits for the $K + 1$ masks, denoted by $\hat{M} := f_{\text{seg}}(x)$, of the same dimension as $M$. The second-stage classifier then processes $\hat{M}$ and returns the predicted class probability $f(x) = f_{\text{cls}}(\hat{M}) = f_{\text{cls}}(f_{\text{seg}}(x))$. The predicted label is given by $\hat{y} := \arg\max_{i \in [C]} f(x)_i$. Fig. 1 visually summarizes our design.

We use DeepLabv3+ (Chen et al., 2018) with ResNet-50 backbone (He et al., 2016) as the segmenter, but our part-based model is agnostic to the choice of segmenter architecture. Additionally, all of the classifiers are designed to be end-to-end differentiable. This facilitates the evaluation process as well making our models compatible with adversarial training.

**Classifier design principles.** We experimented with various classifier architectures, each of which processes the predicted masks differently. Our design criteria were:

1. *Part-based classification*: The classifier should only predict based on the output of the segmenter. It does not see the original image. If the segmenter is correct and robust, the class label can be easily obtained from the masks alone.
2. *Disentangle irrelevant features*: The background is not part of the object being classified so the segmenter should separate it from the foreground pixels. Sometimes, background features could result in spurious correlation (Xiao et al., 2021; Sagawa et al., 2020). Thus, we could simply drop the predicted background pixels or leave it to the second-stage classifier to correctly utilize them. This design choice is explored further in Appendix D.7.
3. *Location-aware*: The second-stage classifier should utilize the *location* and the *size* of the parts, in addition to their existence.

Following these principles, we designed four part-based classifiers, *Downsampled*, *Bounding-Box*, *Two-Headed*, and *Pixel*. The first two perform as well or better than the others, so we focus only on them in the main text of the paper. Appendix C has details on the others.

## 3.2 DOWNSAMPLED PART-BASED MODEL

This model first applies softmax on the predicted mask logits $\hat{M}$ to normalize the masks pixel-wise to a number between 0 and 1. This potentially benefits robustness: if the masks were not normalized, a few pixels could be manipulated to have a very large value and outweigh the benign pixels. Empirically, this softmax doesn't lead to gradient obfuscation (Athalye et al., 2018) (Appendix D.3).

After that, the masks are downsampled to size $4 \times 4$ ($\mathbb{R}^{K \times 4 \times 4}$) by an adaptive average pooling layer before being passed to a tiny neural network with one convolution layer and two fully-connected layers. Fig. 3a illustrates this model. Downsampling maintains coarse-grained spatial information about each part's rough shape and location while compressing high-dimensional masks to a low-dimensional feature vector. This keeps the classifier small, making the part-based model comparable

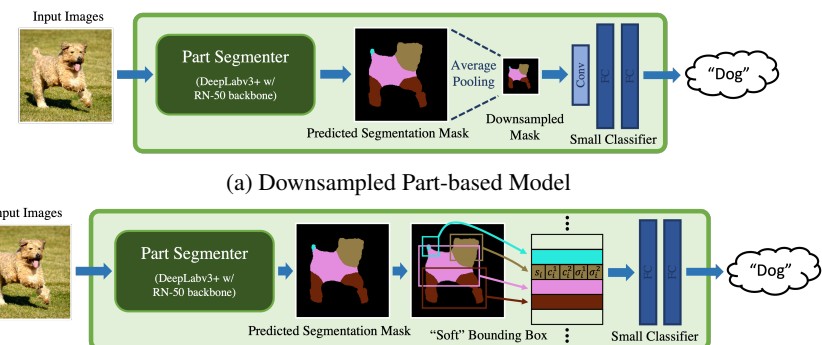

(a) Downsampled Part-based Model

(b) Bounding-Box Part-based Model

Figure 3: Illustration of our two part-based models: (a) downsampled and (b) bounding-box.

to the normal ResNet-50 in size. We find that the particular size of the downsampled mask has little effect on the accuracy of the model (see Appendix D.8 for the comparison).

### 3.3 BOUNDING-BOX PART-BASED MODEL

Similar to the downsampled classifier, the bounding-box classifier also compresses $\hat{M}$ to a lower-dimensional representation, but instead of downsampling, it uses bounding boxes. Specifically, it processes each of the logit segmentation masks, $\hat{M}_i$, into $K$ "soft" bounding boxes, one for each object part, excluding the background channel (see Fig. 3b). Each bounding box is represented by five features: a logit score ($s_i \in [0, 1]$), a centroid ($c_i^1, c_i^2 \in [-1, 1]$) representing the (normalized) 2D coordinates of the center of the bounding box, and a standard deviation ($\sigma_i^1, \sigma_i^2 \in [0, 1]$) capturing the height and the width of the box. We describe how these features are computed below. This gives us a dense feature vector $v = [v_1, \ldots, v_K] \in \mathbb{R}^{5K}$ where $v_i = [s_i, c_i^1, c_i^2, \sigma_i^1, \sigma_i^2] \in \mathbb{R}^5$. Finally, a tiny fully-connected neural network predicts the class label given $v$ and no other information.

Crucially, we ensure that the computation of these features is differentiable to enable effective training as well as reliable evaluation of adversarial robustness. First, we compute a mask $\hat{F}$ that for all foreground pixels. Then, the confidence score for each part mask, $s_i$, is the weighted average of the part logit mask $\hat{M}_i$ over all pixels, weighted by $\hat{F}$.

$$s_i = \frac{\sum_{h=1}^{H} \sum_{w=1}^{W} \hat{M}_i^{(h,w)} \cdot \hat{F}^{(h,w)}}{\sum_{h=1}^{H} \sum_{w=1}^{W} \hat{F}^{(h,w)}} \quad \text{where} \quad \hat{F}^{(h,w)} = \text{Sigmoid}\left(\sum_{k=1}^{K} \hat{M}_k^{(h,w)} - \hat{M}_0^{(h,w)}\right), \quad (1)$$

The other four bounding-box features are computed as follows:

$$c_i^1 = \sum_{h=1}^{H} p_i(h) \cdot h, \qquad \sigma_i^1 = \sqrt{\sum_{h=1}^{H} p_i(h) \cdot (h - c_i^1)^2} \tag{2}$$

$$c_i^2 = \sum_{w=1}^{W} p_i(w) \cdot w, \qquad \sigma_i^2 = \sqrt{\sum_{w=1}^{W} p_i(w) \cdot (w - c_i^2)^2} \tag{3}$$

$$\text{where} \quad p_i(h') = \frac{\sum_{w=1}^{W} \bar{M}_i^{(h',w)}}{\sum_{h=1}^{H} \sum_{w=1}^{W} \bar{M}_i^{(h,w)}}, \quad p_i(w') = \frac{\sum_{h=1}^{H} \bar{M}_i^{(h,w')}}{\sum_{h=1}^{H} \sum_{w=1}^{W} \bar{M}_i^{(h,w)}}, \tag{4}$$

and $\bar{M}^{(h,w)} = \text{Softmax}\left(\hat{M}^{(h,w)}\right)_{[1,\ldots,K]}$ is the softmax mask with the background channel removed.

Note that $p_i(h)$ and $p_i(w)$ can be interpreted as the (normalized) density of the $i$-th object part in row $h$ or column $w$, and $\hat{M}_i^{(h,w)}$ as its mass. Hence, $c_i^1$ and $c_i^2$ are simply the centroid of the $i$-th part. $\sigma_i^1$ and $\sigma_i^2$ measure the spread of mass so we use them as a proxy for the height and the width of the part.

### 3.4 TRAINING LOSSES

**Normal loss.** These part-based models are trained end-to-end with a combined *segmentation-classification* loss, i.e., a weighted sum of two cross-entropy losses, one for the classification task and

one for the pixel-wise segmentation task. A hyperparameter, $c_{\mathrm{seg}} \in [0, 1]$, balances these two losses.

$$L_{\mathrm{normal}}(x,y) = (1 - c_{\mathrm{seg}}) \cdot L_{\mathrm{cls}}(x,y) + c_{\mathrm{seg}} \cdot L_{\mathrm{seg}}(x,y) \tag{5}$$

$$\text{where} \qquad L_{\mathrm{cls}}(x,y) = L_{\mathrm{CE}}\left(f(x), y\right) \tag{6}$$

$$\text{and} \qquad L_{\mathrm{seg}}(x,y) = \frac{1}{(K+1)HW} \sum_{k=0}^{K} \sum_{j=1}^{H \times W} L_{\mathrm{CE}}\left(f_{\mathrm{seg}}(x), M_k^{(j)}\right). \tag{7}$$

**Adversarial loss.** We construct an adversarial version of this loss, that measures susceptibility to adversarial examples. The adversary's goal is to maximize the classification loss (since it is the main task we evaluate on). The same adversarial example $x^*$ generated from the classification loss is also used to compute the segmentation loss.

$$L_{\mathrm{adv}}(x,y) = (1 - c_{\mathrm{seg}}) \cdot L_{\mathrm{cls}}(x^*,y) + c_{\mathrm{seg}} \cdot L_{\mathrm{seg}}(x^*,y) \tag{8}$$

$$\text{where} \qquad x^* = \argmax_{z: \|z-x\|_p \le \epsilon} L_{\mathrm{cls}}(z,y) \tag{9}$$

**TRADES loss.** We combine this with TRADES loss (Zhang et al., 2019) which introduces an extra term, a Kullback–Leibler divergence ($D_{\mathrm{KL}}$) between the clean and the adversarial probability output.

$$L_{\mathrm{TRADES}}(x,y) = (1 - c_{\mathrm{seg}}) \cdot L_{\mathrm{cls}}(x,y) + c_{\mathrm{seg}} \cdot L_{\mathrm{seg}}(x^*,y) + \beta \cdot D_{\mathrm{KL}}\left(f(x), f(x^*)\right) \tag{10}$$

$$\text{where} \qquad x^* = \argmax_{z: \|z-x\|_p \le \epsilon} D_{\mathrm{KL}}\left(f(x), f(z)\right) \tag{11}$$

### 3.5 EXPERIMENT SETUP

#### 3.5.1 DATASET PREPARATION

We demonstrate our part models on three datasets where part-level annotations are available: Part-ImageNet (He et al., 2021), Cityscapes (Meletis et al., 2020), and PASCAL-Part (Chen et al., 2014).

Cityscapes and PASCAL-Part were originally created for segmentation, so we construct a classification task from them. For Cityscapes, we create a human-vs-vehicle classification task. For each human or vehicle instance with part annotations, we crop a square patch around it with some random amount of padding and assign the appropriate class label. PASCAL-Part samples do not require cropping because each image contains only a few objects, so we simply assign a label to each image based on the largest object in that image. To deal with the class imbalance problem, we select only the top five most common classes. Appendix A presents additional detail on the datasets.

#### 3.5.2 NETWORK ARCHITECTURE AND TRAINING PROCESS

ResNet-50 (He et al., 2016) is our baseline. Our part-based models (which use DeepLabv3+ with ResNet-50 backbone) have a similar size to the baseline: our part-based models have 26.7M parameters, compared to 25.6M parameters for ResNet-50. All models are trained with SGD and a batch size of 128, using either adversarial training or TRADES, with 10-step $\ell_\infty$-PGD with $\epsilon = 8/255$ and step size of $2/255$. Training is early stopped according to adversarial accuracy computed on the held-out validation set. All models, both ResNet-50 and part-based models, are pre-trained on unperturbed images for 50 epochs to speed up adversarial training (Gupta et al., 2020).

#### 3.5.3 HYPERPARAMETERS

Since our experiments are conducted on new datasets, we take particular care in tuning the hyper-parameters (e.g, learning rate, weight decay factor, TRADES' $\beta$, and $c_{\mathrm{seg}}$) for both the baseline and our part-based models. For all models, we use grid search on the learning rate $(0.1, 0.05, 0.02)$ and the weight decay $(1 \times 10^{-4}, 5 \times 10^{-4})$ during PGD adversarial training. For the part-based models, after obtaining the best learning rate and weight decay, we then further tune $c_{\mathrm{seg}}$ by sweeping values $0.1, 0.2, \ldots, 0.9$ and report on the model with comparable adversarial accuracy to the baseline. Results for other values of $c_{\mathrm{seg}}$ are included in Section D.2.

For TRADES, we reuse the best hyperparameters obtained previously and sweep a range of the TRADES parameter $\beta$, from 0.05 to 2, to generate the accuracy-robustness trade-off curve. However,

Table 1: Comparison of normal and part-based models under different training methods. Adversarial accuracy is computed with AutoAttack ($\epsilon = 8/255$). For TRADES, we first train a ResNet-50 model with clean accuracy of at least 90%, 96%, and 80% for Part-ImageNet, Cityscapes, and PASCAL-Part, respectively, then we train part-based models with similar or slightly higher clean accuracy.

| Training Method | Models | Part-ImageNet | | Cityscapes | | PASCAL-Part | |
|---|---|---|---|---|---|---|---|
| | | Clean | Adv. | Clean | Adv. | Clean | Adv. |
| PGD (Madry et al., 2018) | ResNet-50 | 74.7 | 37.7 | 79.5 | 68.4 | 47.1 | 37.8 |
| | Downsampled Part Model | 85.6 (↑ 10.9) | 39.4 (↑ 1.7) | 94.8 (↑ 15.3) | 70.2 (↑ 1.8) | 49.6 (↑ 2.5) | 38.5 (↑ 0.7) |
| | Bounding-Box Part Model | 86.5 (↑ 11.8) | 39.2 (↑ 1.5) | 94.2 (↑ 14.7) | 69.9 (↑ 1.4) | 52.2 (↑ 5.1) | 38.5 (↑ 0.7) |
| TRADES (Zhang et al., 2019) | ResNet-50 | 90.6 | 7.7 | 96.7 | 52.5 | 80.2 | 12.6 |
| | Downsampled Part Model | 90.9 (↑ 0.3) | 19.8 (↑ 12.1) | 97.1 (↑ 0.4) | 62.5 (↑ 10.0) | 83.1 (↑ 2.9) | 29.9 (↑ 17.3) |
| | Bounding-Box Part Model | 90.8 (↑ 0.2) | 24.1 (↑ 16.4) | 97.1 (↑ 0.4) | 63.0 (↑ 10.5) | 88.5 (↑ 8.3) | 29.5 (↑ 16.9) |

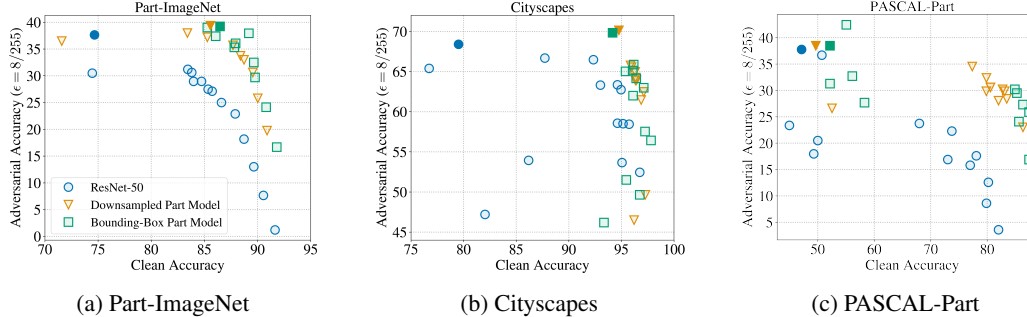

|     (a) Part-ImageNet     |     (b) Cityscapes     |     (c) PASCAL-Part     |

Figure 4: Accuracy and robustness trade-off plots of normal and part-based models trained on (a) Part-ImageNet, (b) Cityscapes, and (c) PASCAL-Part. The filled dots represent PGD adversarial training while the unfilled ones denote TRADES with different values of its parameter $\beta$.

we do not tune $c_{\text{seg}}$ here and keep it fixed at $0.5$ which puts equal weight on the classification and the segmentation losses. The same hyperparameter tuning strategy is used on both the baseline and our part models. We include our code along with the data preparation scripts in the supplementary material. Appendix B contains a detailed description of the experiment.

## 4    ROBUSTNESS EVALUATION

We compare the adversarial robustness and the clean accuracy of the part-based models to the ResNet-50 baseline. We must examine both metrics at the same time since there is a known trade-off between them (Tsipras et al., 2019). We use AutoAttack (Croce & Hein, 2020), a standard and reliable ensemble of attacks, to compute the adversarial accuracy of all models. We also follow the suggested procedures from Carlini et al. (2019) to ensure that our evaluation is free from the notorious gradient obfuscation problem. For further discussion, see Appendix D.3.

Table 1 compares the part-based models to the baseline ResNet-50 under two training methods: PGD adversarial training (Madry et al., 2018) and TRADES (Zhang et al., 2019). For PGD-trained models, both of the part-based models achieve about **3–15 percentage points higher clean accuracy than the baseline** with similar adversarial accuracy. The models trained on Cityscapes show the largest improvement, followed by ones on Part-ImageNet and PASCAL-Part. TRADES allows controlling the tradeoff between clean vs adversarial accuracy, so we choose models with similar clean accuracy and compare their robustness. **The part models outperform the baseline by about 16, 11, and 17 percentage points on Part-ImageNet, Cityscapes, and PASCAL-Part, respectively.** These results show that part-based models significantly improve adversarial robustness.

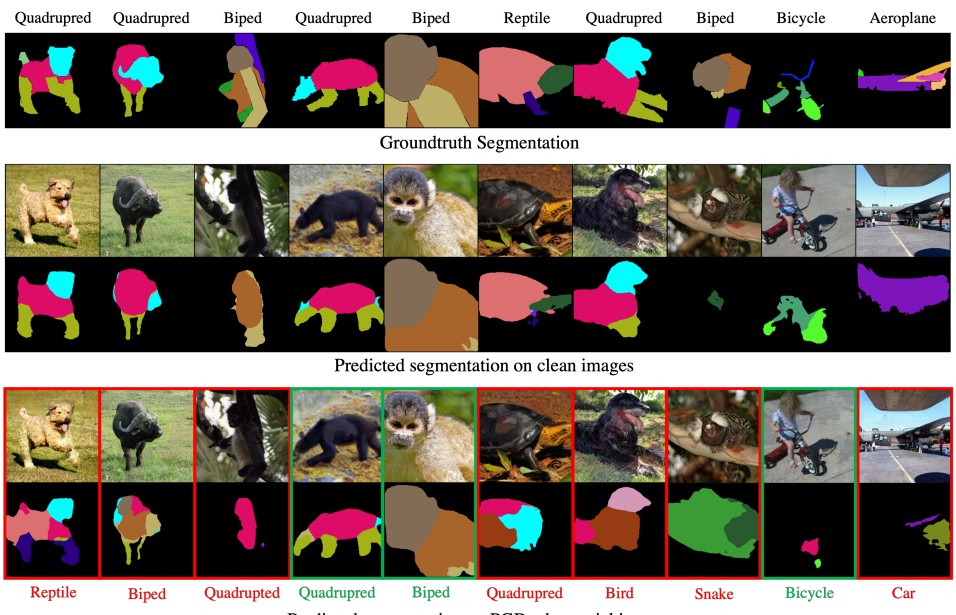

Figure 5: Visualization of the part segmentation predicted by the segmenter of the bounding-box part model adversarially trained on Part-ImageNet. All of the clean samples shown in the second and the third rows are correctly classified. The last two rows show PGD adversarial examples and their predictions. The misclassified (resp. correctly classified) samples are indicated with a red (resp. green) box, and the misclassified class labels are shown below in red (resp. green). The ground-truth labels and segmentation mask can be found on the top row.

Fig. 4 plots the robustness-accuracy trade-off curves for all three datasets, generated by sweeping the TRADES hyperparameter $\beta$ (see Section 3.5.3). Our part-based models are closer to the top-right corner of the plot, indicating that they outperform the baseline on both clean and adversarial accuracy.

Fig. 5 shows ten randomly chosen test samples from Part-ImageNet along with their predictions from the adversarially trained bounding-box part model, with and without the attack. Most of the part-based models, including this one, achieve above $80\%$ pixel-wise segmentation accuracy on clean samples and about $70\%$ on adversarial ones. Successful attacks can change most, but not all, foreground pixels to the wrong class, but the shape and foreground-vs-background prediction for each part remains correct; the attack changes only the predicted class for each part. This suggests that part-based models may learn shape features that are more difficult to manipulate, an observation that aligns with our quantitative results on shape-vs-texture bias in Section 5.1. We suspect the robustness of these part shapes might account for the model's improved robustness.

**Attacking the segmenter model.** To ensure that we evaluate our models with the strongest attack possible, we come up with two additional attacks that target the segmenter. First is the *single-staged attack* which optimizes a combination of the classification and the segmentation losses as in Eqn. 8. The second attack is the *two-staged attack* where the first stage attacks the segmenter alone to produce the worst-case mask. This step generates "guiding samples" which are then used to initialize the second stage that attacks the part model end-to-end. For this attack, we experiment with four variations that differ in how the target masks are chosen in the first stage. We find that the single-stage attack is *always* worse than the normal PGD attack. A small subset of the two-stage attacks performs better than PGD, but all of them are worse than AutoAttack. For more details, see Appendix D.3.

# 5 UNDERSTANDING THE PART-BASED MODELS

## 5.1 EVALUATING NON-ADVERSARIAL ROBUSTNESS

Part-based models improve adversarial robustness, but what about robustness to non-adversarial distribution shift? We evaluate the models on three scenarios: common corruptions, foreground-vs-background spurious correlation, and shape-vs-texture bias. We generate benchmarks from Part-ImageNet following the same procedure as ImageNet-C (Hendrycks & Dietterich, 2019) for common

Table 2: Accuracy on the common corruption benchmark. We report a 95% confidence interval across different random seeds for training.

| Model | Corruption Robustness |
|---|---|
| ResNet-50 | $82.3 \pm 1.6$ |
| Downsampled Part Model | $85.5 \pm 0.8$ |
| Bounding-Box Part Model | $\mathbf{85.8} \pm 0.7$ |

Table 3: Accuracy on the background/foreground spurious correlation benchmark, with 95% CI across different random seeds.

| Model | Spurious Correlation |
|---|---|
| ResNet-50 | $58.6 \pm 4.2$ |
| Downsampled Part Model | $\mathbf{65.1} \pm 0.8$ |
| Bounding-Box Part Model | $\mathbf{65.1} \pm 2.1$ |

Table 4: Accuracy on the shape-vs-texture bias benchmark. We report a 95% confidence interval across 10 different random seeds for training. Higher accuracy is better, suggesting that the model is less dependent on the texture features and more biased toward robust shape features.

| Model | Shape-vs-Texture |
|---|---|
| ResNet-50 | $40.6 \pm 1.8$ |
| Downsampled Part Model | $44.7 \pm 2.6$ |
| Bounding-Box Part Model | $\mathbf{45.7} \pm 2.7$ |

Table 5: Clean and adversarial accuracy of part-based models trained with and without the part segmentation labels compared to the ResNet-50 baseline. The improvement from the segmentation labels is highlighted.

| Models | Seg. Labels? | Clean | Adv. |
|---|---|---|---|
| ResNet-50 | N/A | 74.7 | 37.7 |
| Downsampled Part Model | No | 76.9 | 39.6 |
|  | Yes | 85.6 | 39.4 |
|  |  | ($\uparrow 8.7$) | ($\downarrow 0.2$) |
| Bounding-Box Part Model | No | 78.1 | 39.9 |
|  | Yes | 86.5 | 39.2 |
|  |  | ($\uparrow 8.4$) | ($\downarrow 0.7$) |

corruptions, ImageNet-9 (Xiao et al., 2021) for foreground-vs-background spurious correlation, and Stylized ImageNet (Geirhos et al., 2019) for shape-vs-texture bias. For the common corruptions, the benchmark is composed of 15 corruption types and five severity levels. The spurious correlation benchmark is generated from a subset of foreground ("Only-FG") and background ("Only-BG-T") of ImageNet-9, filtering out classes not present in Part-ImageNet. Each foreground image is paired with a randomly chosen background image of another class. For shape-vs-texture bias, the data are generated by applying styles/textures using neural style transfer.

We train a ResNet-50 model and two part-based models using conventional training (not adversarial training) on clean Part-ImageNet samples. We tune the hyperparameters as described in Section 3.5.3. For each benchmark, the best-performing set of hyperparameters is used to train 10 randomly initialized models to compute the confidence interval.

**On all of the benchmarks, the part-based models outperform the baseline by 3–7 percentage points** (see Tables 2, 3, and 4). The improvement over the ResNet-50 baseline is statistically significant (two-sample $t$-test, $p$-values below $10^{-6}$). We note that these robustness gains do *not* come at a cost of clean accuracy as the clean accuracy of our part models is about 1% higher on average than that of the ResNet-50. This suggests that part-based models are more robust to common corruptions, better disentangle foreground and background information, and have higher shape bias compared to typical convolutional neural networks.

## 5.2 EFFECTS OF PART SEGMENTATION LABELS VS ARCHITECTURE

Where does the robustness improvement come from? Does it come from the additional information provided by part annotations, or from the new architecture we introduce?

To answer these questions, we train part-based models on Part-ImageNet without using the part segmentation labels while keeping the model architecture and hyperparameters fixed (i.e., setting $L_{seg}$ in Eqn. 5 to zero). We found that most of the improvement comes from the additional supervision provided by part annotations. In particular, the architecture provides 2–4 percentage points of improvement over ResNet-50, while the additional supervision provides another 8–9 points of improvement in clean accuracy (see Table 5). This experiment confirms that most of the gain comes from the additional information provided by fine-grained segmentation labels.

We also extend this ablation study and consider other backbone architectures. We replace ResNet-50 with EfficientNet-B4 and ResNeXt-50-32x4d. We find that the part-level supervision still improves the model's accuracy and robustness by a large margin (see Appendix D.8.1 and Table 18).

Table 6: Comparison of accuracy of part models trained using different types of auxiliary labels. The part bounding-box and centroid models are PGD adversarially trained. We select the part segmentation model with similar accuracy from Section 4 for comparison.

| Types of Labels | Part-ImageNet | | Cityscapes | | PASCAL-Part | |
|---|---|---|---|---|---|---|
| | Clean | Adv. | Clean | Adv. | Clean | Adv. |
| Segmentation | 85.6 | 39.4 | 94.8 | 70.2 | 77.3 | 34.5 |
| Bounding Boxes | 84.1 | 39.7 | 95.4 | 69.1 | 66.2 | 33.5 |
| Centroids | 82.6 | 39.7 | 94.0 | 70.9 | 62.9 | 33.5 |
| ResNet-50 | 74.7 | 37.7 | 79.5 | 68.4 | 54.0 | 29.1 |

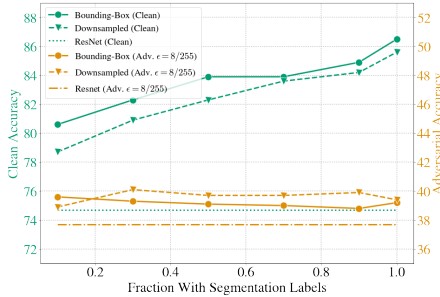

Figure 6: Performance of the part models when only a fraction of training samples are accompanied by a segmentation label.

### 5.3 TRAINING WITH FEWER PART SEGMENTATION LABELS

The main limitation of our approach is the extra labeling cost to obtain part segmentation labels. We investigate how the performance of our models changes when fewer part annotations are available. We train part models with the same number of training samples and class labels but a reduced number of segmentation labels, so some (10–90%) of training samples have both class and segmentation labels while the rest have *only* the class label. As expected, the clean accuracy degrades when the model receives less part-level supervision; see Fig. 6. Adversarial accuracy, however, remains more or less the same, and all of the part models still outperform the ResNet-50.

Given this observation, we attempt to reduce the reliance on the part segmentation labels. Surprisingly, we find that **simple pseudo-labeling can reduce the required training labels down by 90%!** Using labels generated by another segmentation model results in part models with almost the same accuracy and robustness as the fully-supervised models. See Appendix D.6 and Table 15 for more details.

### 5.4 ALTERNATIVES TO PART SEGMENTATION LABELS

We additionally explore two labeling strategies for reducing labelling costs: (1) bounding box segmentations for each part, or (2) keypoints or centroids for each part (Fig. 10, Appendix D.4).[1] These annotations provide less precise spatial information about each part but are much faster to label. Bounding-box labels are nearly as effective as segmentation masks on Part-ImageNet and Cityscapes (within ~1% difference in accuracy; see Table 6). However, the difference is much larger on PASCAL-Part where the clean accuracy is 11% lower. Models trained on centroid labels perform slightly worse than the ones trained on bounding-box labels, which is unsurprising as centroids are even more coarse-grained. Nonetheless, all part models trained on any kind of part label still outperform the ResNet-50 baseline. We hope our work draws attention to the opportunity for stronger robustness through rich supervision and stimulates research into reducing the labeling cost.

We also conduct an ablation study by replacing part segmentation labels with *object* segmentation labels. Models trained on object segmentation labels are less effective than ones trained on part labels (Appendix D.5). Even though models trained with object-level labels outperform the baseline, this implies that part-level annotation is still important.

## 6 CONCLUSION

In this work, we propose a new approach to improve adversarial training by leveraging a richer learning signal. We materialize this concept through part-based models that are trained simultaneously on object class labels and part segmentation. Our models outperform the baseline, improving the accuracy-robustness trade-off, while also benefiting non-adversarial robustness. This work suggests a new direction for work on robustness, based on richer supervision rather than larger training sets.

---

[1] Here, we refer to the labels provided for training. This should not be confused with the architecture of the Bounding-Box Part Model.

ACKNOWLEDGEMENTS

The authors would like to thank Vikash Sehwag and Jacob Steinhardt for their feedback on the paper. We also thank the anonymous ICLR reviewers for their helpful comments and suggestions. This research was supported by the Hewlett Foundation through the Center for Long-Term Cybersecurity (CLTC), by the Berkeley Deep Drive project, and by generous gifts from Open Philanthropy and Google Cloud Research Credits program under Award GCP19980904. Jacob Steinhardt also generously lent us the computing resources used in this research.

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

## A    DATASETS

**Part-ImageNet.**    Proposed by He et al. (2021), this dataset is a subset of ImageNet-1K where the 158 of the original classes are grouped into 11 coarse-grained classes, e.g., "Quadruped," "Biped," "Reptile," etc. Each object is accompanied by pixel-wise annotation of 2–5 parts. For instance, a quadruped may have up to four segmentation masks for its head, body, feet, and tail. The dataset is originally proposed for part segmentation or part discovery tasks and is publicly available to download.[2] We note that the Part-ImageNet dataset splits the data by their original ImageNet-1K classes, i.e., 109, 19, and 30 classes for training, validation, and test sets, respectively. This allows one to measure generalization across sub-population under the same group. However, our focus is different; we want to evaluate the robustness under a similar setting to CIFAR-10 whose samples are split i.i.d. Hence, for this paper, we ignore the original ImageNet class and re-partition the dataset randomly, independent of its original class. The Part-ImageNet dataset has 24,095 samples in total.

**Cityscapes.**    The Cityscapes dataset is a driving-oriented image dataset whose data were collected from a dashboard camera (Cordts et al., 2016). We use the part-aware panoptic annotations on Cityscapes from Meletis et al. (2020) to create our classification dataset. The Cityscapes dataset is available under a non-commercial license[3], and the annotation is available under Apache-2.0.[4] Five kinds of objects are part-annotated, and we group them into two classes. Specifically, "person" and "rider" are grouped as "human," and "car," "truck," and "bus" as "vehicle." We use the same part labels as Meletis et al. (2020) where humans are annotated with "torso," "head," "arms," and "legs," and vehicles with "windows," "wheels," "lights," "license plate," and "chassis."

Since the samples in Cityscapes are wide-angle photos containing numerous objects, we crop each annotated object out to create a classification dataset. In particular, we crop each patch into a square and then add a small amount of extra random padding (0–10% of the image size). Additionally, we also filter out small objects that have the total area, determined from the segmentation mask of the entire object, less than 1000 pixels. After filtering, we are left with 29,728 samples in the dataset.

**PASCAL-Part.**    The PASCAL-Part dataset (Chen et al., 2014) provides part-aware segmentation annotation of the PASCAL VOC (2010) dataset (Everingham et al., 2010) which is an object recognition and detection dataset. Both the annotations and the original dataset are available to the public.[5] The original PASCAL-Part dataset comprises 20 classes, but most of them have 500 or fewer samples. To ensure that we have a sufficient number of samples per class and avoid the class imbalance problem, we opt to select only the top-five most common classes: "aeroplane," "bird," "car," "cat," and "dog." In PASCAL-Part, the parts are annotated in a more fine-grained manner, compared to the other two datasets. For example, the legs of a dog are labeled as front or back and left or right. To make the number of parts per object manageable and comparable to the other two datasets, we group multiple parts of the same type together, e.g., all legs are labeled as "legs." Our PASCAL-Part dataset has 3,662 samples in total.

We also emphasize that we do not use a common benchmark dataset such as CIFAR-10 since it is not part-annotated and is too low-resolution to be useful in practice. The datasets we use are more realistic and have much higher resolution. For training and testing the models, we use the same preprocessing and data augmentation as commonly used for the ImageNet dataset. Specifically, the training samples are randomly cropped and resized to $224 \times 224$ pixels, using PyTorch's `RandomResizedCrop` function with the default hyperparameters, and applied a random horizontal flip. Test and validation samples are center cropped to $256 \times 256$ pixels and then resize to $224 \times 224$ pixels.

---

[2] https://github.com/tacju/partimagenet.

[3] https://www.cityscapes-dataset.com/license/

[4] For the Cityscape dataset, https://www.cityscapes-dataset.com/, and for the annotation, https://github.com/pmeletis/panoptic_parts/tree/master/panoptic_parts/cityscapes_panoptic_parts/dataset_v2.0.

[5] PASCAL VOC and its license can be found at http://host.robots.ox.ac.uk/pascal/VOC/voc2010/, and for PASCAL-Part, see https://roozbehm.info/pascal-parts/pascal-parts.html.

## B    DETAILED EXPERIMENT SETUP

Here, we provide information regarding the model implementation in addition to Section 3.5. All models are adversarially trained for 50 epochs. To help the training converge faster, we also pre-train every model on clean data for 50 epochs before tuning on adversarial training as suggested by Gupta et al. (2020). We save the weights with the highest accuracy on the held-out validation data which does not overlap with the training or the test set. We use the cosine annealing schedule to adjust the learning rate as done in Loshchilov & Hutter (2017). Our experiments are conducted on Nvidia GeForce RTX 2080 TI and V100 GPUs.

To evaluate all the models, we rely on both the strong ensemble AutoAttack and the popular PGD attack. However, the AutoAttack is always stronger than the PGD attack in all of the cases we experiment with so we only report the adversarial accuracy corresponding to the AutoAttack in the main paper. AutoAttack comprises four different attacks: adaptive PGD with cross-entropy loss (`apgd-ce`), targeted adaptive PGD with DLR loss (`apgd-t`), targeted FAB attack (`fab-t`), and Square attack (`square`) (Croce & Hein, 2020). However, since the DLR loss requires that there are four or more classes, we have to adapt the AutoAttack on the Cityscapes dataset which has two classes. As a result, we use only three attacks and remove the targeted ones which leave adaptive PGD with cross-entropy loss (`apgd-ce`), FAB attack (`fab`), and Square attack (`square`). We use the default hyperparameters for all of the attacks in AutoAttack. For the PGD attack, we use a step size of 0.001 with 100 iterations and five random restarts.

## C    DESCRIPTIONS AND RESULTS ON THE REMAINING CLASSIFIER ARCHITECTURE

### C.1    TWO-HEADED PART MODEL

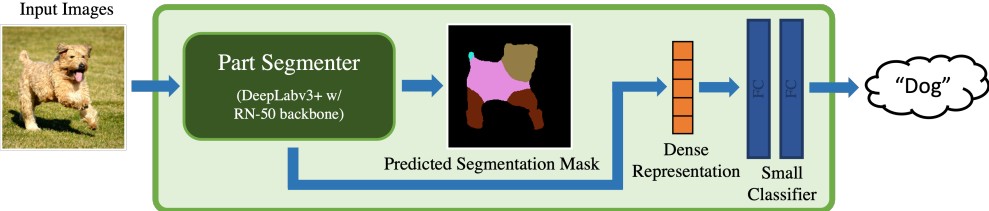

Figure 7: Diagram of the two-headed part model.

The two-headed part model uses a similar architecture to multi-task models with multiple heads. Here, there are two heads, one for segmentation and one for classification, sharing the same dense representation from the bottleneck layer of DeepLabv3+, as illustrated by Fig. 7. It is important to note that the two-headed part model does not explicitly use the predicted segmentation masks in classification. Instead, the classifier only sees the dense representation that will later be turned into the segmentation mask by the remaining layers of the segmenter. From an information-theoretic standpoint, the classifier of the two-headed part model should receive equal or more information than the classifier in the bounding box or the downsampled part model. The difference is that this information is represented as dense vectors in the two-headed part model. However, in the other two models, the information is more human-interpretable and more compressed.

### C.2    PIXEL PART MODEL

The pixel part model is arguably the simplest among all of our part-based models. It does not use a small neural network classifier and involves only two simple steps. First, for each pixel, it sums together the part logits belonging to the same object class. In other words, the part segmentation mask is turned into the object segmentation mask, i.e., $\mathbb{R}^{(K+1) \times H \times W} \to \mathbb{R}^{C \times H \times W}$ where $K$ and $C$ are the numbers of parts and classes, respectively. Then, the object scores are averaged across all pixels in the segmentation mask to obtain the final class logits. This model is summarized in Fig. 8. It is also possible to treat the pixel part model as a specific case of the downsampled one where the

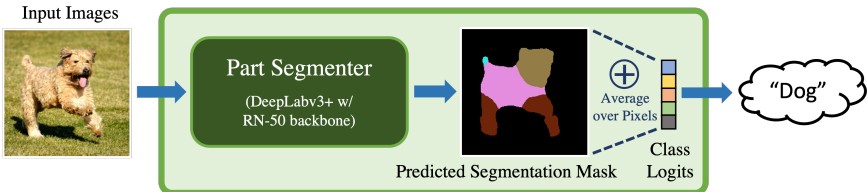

Figure 8: Diagram of the pixel part model.

Table 7: Clean and adversarial accuracy of *the ResNet-50 baseline* obtained over our hyperparameter sweep on Part-ImageNet.

| Training Method | Learning Rate | Weight Decay | TRADES $\beta$ | Clean | AutoAttack | PGD |
|---|---|---|---|---|---|---|
| Normal | 0.1 | $5 \times 10^{-4}$ | N/A | 92.9 | 0.0 | 0.0 |
| PGD | 0.1 | $5 \times 10^{-4}$ | N/A | 74.7 | 37.7 | 43.3 |
| | | $1 \times 10^{-4}$ | N/A | 69.2 | 36.5 | 40.7 |
| | 0.05 | $5 \times 10^{-4}$ | N/A | 76.4 | 36.6 | 42.2 |
| | | $1 \times 10^{-4}$ | N/A | 74.8 | 34.4 | 40.3 |
| | 0.02 | $5 \times 10^{-4}$ | N/A | 73.0 | 33.6 | 39.6 |
| | | $1 \times 10^{-4}$ | N/A | 70.5 | 32.0 | 37.6 |
| TRADES | 0.1 | $5 \times 10^{-4}$ | 0.05 | 91.6 | 1.2 | 2.2 |
| | | | 0.1 | 90.6 | 7.7 | 10.3 |
| | | | 0.15 | 89.6 | 13.0 | 16.0 |
| | | | 0.2 | 88.7 | 18.2 | 21.5 |
| | | | 0.3 | 87.9 | 22.9 | 26.0 |
| | | | 0.4 | 86.6 | 25.0 | 28.5 |
| | | | 0.5 | 85.7 | 27.1 | 29.8 |
| | | | 0.6 | 85.4 | 27.5 | 31.5 |
| | | | 0.7 | 84.7 | 29.0 | 32.2 |
| | | | 0.8 | 84.0 | 29.0 | 32.5 |
| | | | 0.9 | 83.8 | 30.6 | 34.2 |
| | | | 1.0 | 83.4 | 31.2 | 35.1 |
| | | | 2.0 | 74.4 | 30.5 | 34.9 |

convolution layer with a kernel size of $1 \times 1$ mimics the first step, and the classifier represents the average function in the second.

Importantly, averaging the logits across pixels means that the spatial information is ignored completely in the classification process. This eventually results in a minor reduction in the accuracy compared to the downsampled or the bounding-box model as shown in Appendix D.9 and Table 19. We do not recommend this model in practice, and it partially serves as an ablation study in our work.

# D    ADDITIONAL ROBUSTNESS RESULTS

## D.1    HYPERPARAMETER SWEEP RESULTS

In this section, we include detailed results from our hyperparameter sweep on the ResNet-50 baseline (Table 7), the downsampled (Table 8), and the bounding-box part models (Table 9). The results suggest that all of the adversarially trained models are, to some degree, sensitive to the training hyperparameters, e.g., learning rate and weight decay. Nevertheless, the best setting is rather consistent across most of the models as well as the datasets, i.e., a learning rate of $0.1$ and weight decay of $5 \times 10^{-4}$.

Table 8: Clean and adversarial accuracy of *the downsampled part models* obtained over our hyperparameter sweep on Part-ImageNet.

| Training Method | Learning Rate | Weight Decay | $c_{\text{seg}}$ | TRADES $\beta$ | Clean | AutoAttack | PGD |
|---|---|---|---|---|---|---|---|
| Normal | 0.1 | $5 \times 10^{-4}$ | 0.5 | N/A | 95.2 | 0.0 | 0.0 |
| PGD | 0.1 | $5 \times 10^{-4}$ | 0.5 | N/A | 83.9 | 39.9 | 45.3 |
| | | $1 \times 10^{-4}$ | 0.5 | N/A | 79.1 | 39.6 | 45.8 |
| | 0.05 | $5 \times 10^{-4}$ | 0.5 | N/A | 85.1 | 38.8 | 44.7 |
| | | $1 \times 10^{-4}$ | 0.5 | N/A | 82.3 | 37.5 | 43.7 |
| | 0.02 | $5 \times 10^{-4}$ | 0.5 | N/A | 80.4 | 36.9 | 43.4 |
| | | $1 \times 10^{-4}$ | 0.5 | N/A | 82.3 | 35.1 | 42.4 |
| TRADES | 0.1 | $5 \times 10^{-4}$ | 0.5 | 0.05 | 90.9 | 19.8 | 23.8 |
| | | | | 0.1 | 90.0 | 25.8 | 29.5 |
| | | | | 0.2 | 89.6 | 30.6 | 34.7 |
| | | | | 0.3 | 88.7 | 33.0 | 37.5 |
| | | | | 0.4 | 88.4 | 33.7 | 37.7 |
| | | | | 0.5 | 87.7 | 35.7 | 40.0 |
| | | | | 0.8 | 85.3 | 37.2 | 41.2 |
| | | | | 1.0 | 83.4 | 38.0 | 42.2 |

Table 9: Clean and adversarial accuracy of *the bounding-box part models* obtained over our hyperparameter sweep on Part-ImageNet.

| Training Method | Learning Rate | Weight Decay | $c_{\text{seg}}$ | TRADES $\beta$ | Clean | AutoAttack | PGD |
|---|---|---|---|---|---|---|---|
| Normal | 0.1 | $5 \times 10^{-4}$ | 0.5 | N/A | 95.4 | 0.0 | 0.0 |
| PGD | 0.1 | $5 \times 10^{-4}$ | 0.5 | N/A | 83.1 | 37.0 | 43.7 |
| | | $1 \times 10^{-4}$ | 0.5 | N/A | 84.4 | 39.5 | 45.2 |
| | 0.05 | $5 \times 10^{-4}$ | 0.5 | N/A | 86.2 | 37.7 | 43.6 |
| | | $1 \times 10^{-4}$ | 0.5 | N/A | 83.1 | 37.5 | 43.2 |
| | 0.02 | $5 \times 10^{-4}$ | 0.5 | N/A | 84.1 | 36.0 | 42.3 |
| | | $1 \times 10^{-4}$ | 0.5 | N/A | 81.6 | 37.1 | 43.3 |
| TRADES | 0.1 | $5 \times 10^{-4}$ | 0.5 | 0.05 | 91.8 | 16.7 | 19.4 |
| | | | | 0.1 | 90.8 | 24.1 | 27.5 |
| | | | | 0.2 | 89.8 | 29.7 | 33.5 |
| | | | | 0.3 | 89.6 | 32.5 | 36.2 |
| | | | | 0.4 | 89.2 | 38.0 | 38.0 |
| | | | | 0.5 | 87.8 | 35.3 | 39.3 |
| | | | | 0.6 | 87.9 | 36.1 | 40.1 |
| | | | | 0.8 | 86.1 | 37.4 | 41.5 |
| | | | | 1.0 | 85.3 | 39.0 | 43.0 |

## D.2 EFFECTS OF THE $c_{\text{seg}}$ HYPERPARAMETER

To test the effect of the segmentation loss, we train multiple part-based models with $c_{\text{seg}}$ varied from 0 to 1. With $c_{\text{seg}}$ closer to 1, the loss function prioritizes the pixel-wise segmentation accuracy. With $c_{\text{seg}}$ closer to 0, less emphasis is put on the accuracy of segmentation masks. Fig. 9 shows the accuracy with respect to different $c_{\text{seg}}$ values for both downsampled and bounding-box part models. It is, however, inconclusive whether the smaller or the larger value of $c_{\text{seg}}$ is most preferable in this case. There is a vague trend that larger $c_{\text{seg}}$ improves the clean accuracy but reduces the adversarial accuracy, exhibiting some form of trade-off. This overall trend can be explained by the fact that smaller $c_{\text{seg}}$ places more weight on the adversarial classification loss and hence, improves the robustness.

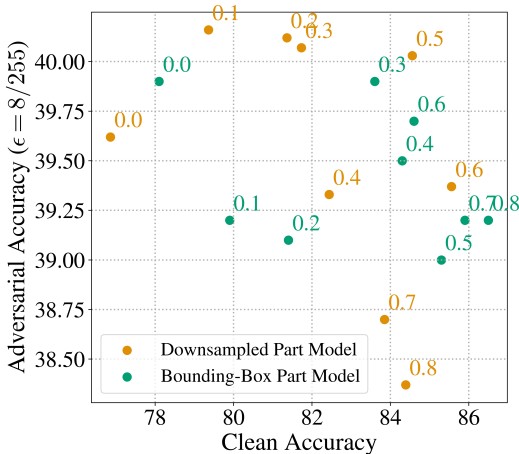

Figure 9: Clean and adversarial accuracy of the downsampled (orange) and the bounding-box (green) part models trained on Part-ImageNet. The number on the top right of each data point indicates the value of $c_{seg}$ that model is trained with. All models are trained with a learning rate of $0.1$ and weight decay of $5 \times 10^{-4}$.

Table 10: Adversarial accuracy of our part-based models at different values of $\epsilon$. This table shows that the adversarial accuracy does reach zero as $\epsilon$ becomes larger which confirms that our part models are unlikely to experience the gradient obfuscation.

| Datasets | Part-Based Models | Adversarial Accuracy | | | |
|---|---|---|---|---|---|
| | | $\epsilon = 8/255$ | $\epsilon = 16/255$ | $\epsilon = 24/255$ | $\epsilon = 32/255$ |
| Part-ImageNet | Downsampled | 39.4 | 13.6 | 3.5 | 1.1 |
| | Bounding-Box | 39.2 | 12.6 | 3.9 | 1.7 |
| Cityscapes | Downsampled | 70.2 | 24.3 | 2.8 | 0.4 |
| | Bounding-Box | 69.9 | 16.6 | 0.9 | 0.0 |
| PASCAL-Part | Downsampled | 38.5 | 24.8 | 8.3 | 1.8 |
| | Bounding-Box | 38.5 | 20.1 | 4.3 | 0.7 |

### D.3 OPTIMALITY OF THE ATTACKS

**Gradient Obfuscation.** We do not believe our models suffer from gradient obfuscation. First, our models do not use any non-differentiable operations or randomization; they use only standard neural network layers.

Second, we conduct a sanity check suggested by Carlini et al. (2019) by making sure that a simple PGD attack can reduce the accuracy close to zero when the perturbation norm increases. Our new experiment, reported in Table 10, confirms this as the adversarial accuracy of our part-based models does drop to $< 2\%$ at $\epsilon = 32/255$.

Third, we have also experimented with decision-based black-box attacks that do not rely on gradient information. We use AutoAttack (Croce & Hein, 2020), which incorporates Square Attack (Andriushchenko et al., 2020) which does not rely on gradients and only uses the output scores. We also use the state-of-the-art $\ell_\infty$-attack, RayS (Chen & Gu, 2020), to evaluate our Downsampled part model on the Part-ImageNet dataset. RayS manages to reduce the accuracy to 71.0 (at 10k steps), which is still much higher than that achieved by the PGD attack and AutoAttack (45.4 and 39.4). This confirms that the non-gradient attacks are not better than the gradient-based ones, suggesting that there is no gradient obfuscation problem.

**Single-staged attack.** We have experimented with multiple ways to attack the part models, including attempts to fool the segmenter by using both losses in the attack objective. However, these alternatives

Table 11: Effects of the $c_{\text{seg}}$ parameter in the loss function of PGD attack on the Downsample part model trained on Part-ImageNet. We emphasize that this is the value of $c_{\text{seg}}$ used during the evaluation attack, not during adversarial training.

| Values of $c_{\text{seg}}$ in PGD Attack | PGD Accuracy |
|---|---|
| 0 (normal PGD) | 45.4 |
| 0.1 | 45.9 |
| 0.3 | 48.0 |
| 0.5 | 50.4 |
| 0.7 | 53.7 |
| 0.9 | 57.5 |

Table 12: Adversarial accuracy measured by the two-staged attack on our part-based models compared to PGD and AutoAttack (AA). "MC" denotes the most-confident strategies.

| Datasets | Part-Based Models | Adversarial Accuracy | | | | | |
|---|---|---|---|---|---|---|---|
| | | PGD | AA | Untargeted | Random | MC (Random) | MC (Sorted) |
| Part-ImageNet | Downsampled | 45.4 | 39.4 | 45.1 | 44.0 | 47.5 | 47.5 |
| | Bounding-Box | 45.7 | 39.2 | 45.3 | 43.3 | 47.3 | 53.4 |
| Cityscapes | Downsampled | 73.8 | 70.2 | 75.4 | 75.5 | 75.4 | 75.5 |
| | Bounding-Box | 73.4 | 69.9 | 74.7 | 74.8 | 74.7 | 74.6 |
| PASCAL-Part | Downsampled | 40.6 | 38.5 | 40.3 | 39.9 | 44.6 | 44.6 |
| | Bounding-Box | 40.6 | 38.5 | 40.6 | 41.0 | 43.9 | 43.2 |

actually decrease the attack success rate. In our experiments, using only classification loss always yields the strongest attack.

In particular, we consider PGD attack with an objective that is a linear combination of the classification loss and the segmentation loss, i.e., $L = (1 - c_{\text{seg}})L_{\text{clf}} + c_{\text{seg}}L_{\text{seg}}$, as in Eqn. 7. Table 11 reports the adversarial accuracy under this attack with varying values of $c_{\text{seg}}$. This shows that using the segmentation loss does not improve the attack. In fact, a larger $c_{\text{seg}}$ (more weight on the segmentation loss) actually results in a worse attack.

**Two-staged attack.** Since we previously found that optimizing over both losses at the same time results in a worse attack, we separate the attack into two stages and make sure that the second stage only optimizes over the classification loss. The difference now lies in the first stage which we use to generate a "guiding sample" to initialize the second attack by focusing on fooling the segmenter first. We experiment with four strategies for the first-stage attack:

1. *Untargeted*: Maximize the loss of the segmenter directly with an untargeted PGD.

2. *Random*: Pick a random target mask from a random incorrect class and run PGD to fool the segmenter into predicting this target mask.

3. *Most-confident (random)*: Similar to the random strategy, but instead of sampling from a random class, only sample target masks from the most-confident class predicted by the part model, excluding the ground-truth class.

4. *Most-confident (sorted)*: Similar to the most-confident (random) strategy, but instead of randomly choosing the masks, we run each mask in the test set through the classifier and choose the ones that the model assigns the highest score/confidence to the target class.

We note that similarly to PGD, we repeat all the two-staged attacks five times with different random seeds and select only the best out of five. This means that the first stage of the attacks uses five different target masks, apart from the untargeted strategy, and produces five different initialization points. Table 12 demonstrates that the two-staged attacks are about as effective as the normal PGD. The untargeted and the random strategies usually perform the best and can be slightly (∼1% lower

Table 13: Comparison of the clean and the adversarial accuracies of the part models with and without adversarial training.

| Models | Adv. Train | Class Adv. Acc. | Seg. Adv. Acc. |
|---|---|---|---|
| Downsampled part model | N | 34.9 | 9.6 |
| | Y | 60.9 | 62.6 |
| Bounding-Box part model | N | 30.5 | 7.8 |
| | Y | 64.4 | 65.5 |

adversarial accuracy) better than the normal PGD attack. Nevertheless, no attack beats AutoAttack in any setting. This suggests that it is likely sufficient to use the AutoAttack alone for evaluation.

**Why is this attack ineffective when Xie et al. (2017) have shown that it is possible to attack segmentation and detection models?** The answer to this lies in the fact that our segmenter has been adversarially trained (end-to-end together with the classifier) whereas the models used in Xie et al. (2017) are only normally trained. To confirm this, we run PGD attacks on the segmenter part of our part models. Table 13 shows that without adversarial training, it is easy to attack the part model and reduce the segmentation accuracy to under 10% (the right-most column). This is in line with Xie et al. (2017). On the other hand, once adversarial training is used, we have a much more robust segmenter with over 60% adversarial accuracy. Since our segmenter is robust, the part model as a whole is also robust.

### D.4 CHEAPER FORMS OF AUXILIARY LABELS

The main limitation of our approach is the need for part segmentation labels. Our primary goal in this paper is to demonstrate that it is possible to achieve significant improvements in robust accuracy using additional supervision. This result is particularly important since progress in the field has somewhat stagnated, and recent improvements through more training samples show diminishing returns (Gowal et al., 2021b). It is an open question whether the most cost-effective way to gain robustness is with more training samples or with richer supervision; our paper provides evidence for the first time that richer (segmentation-like) labels might be a cost-effective route to stronger robustness. We hope our findings will stimulate follow-on research that explores how to achieve these benefits as cheaply as possible. That said, we have tried a few approaches to reduce the labeling cost as already mentioned in Section 5.4. Here, we expand on the experiments that use the bounding-box labels and the centroid labels.

**Bounding-box labels.** The part bounding boxes are generated directly from the part segmentation by drawing a tight box around all the pixels that belong to each part. Fig. 10 provides examples of the bounding box labels from the Part-ImageNet dataset. We want to keep the segmenter unchanged so we train the Downsampled part models with unmodified $L_{\text{seg}}$, as described in Eqn. 7, on the new bounding-box labels. We note that our bounding-box labels are still pixel-wise masks unlike the typical bounding boxes used in the object detection task. In practice, it is likely more efficient to replace the segmenter with an object detection model that outputs bounding boxes directly.

**Centroid labels.** Similarly to the bounding boxes, the centroid labels are also directly derived from the segmentation mask. We go through the same calculation in Eqn. 2 to generate the centroids from the ground-truth, instead of predicted, segmentation masks. Here, we train the bounding-box part model on the centroid labels, but instead of calculating the segmentation loss, we compute the loss directly on the dense features excluding the standard deviations. More precisely, the loss function, $L_{\text{cen}}$, can be written as follows:

$$L_{\text{cen}} = \frac{1}{K} \sum_{k=1}^{K} \left[ (c_k^1(f_{\text{seg}}(x)) - c_k^1(M_k))^2 + (c_k^2(f_{\text{seg}}(x)) - c_k^2(M_k))^2 \right] \quad (12)$$

$$+ L_{\text{CE}} \left( \frac{\sum_{k \in c} \sum_{j=1}^{H \times W} f_{\text{seg}}(x)}{\sum_{c=1}^{C} \sum_{k \in c} \sum_{j=1}^{H \times W} f_{\text{seg}}(x)}, y \right). \quad (13)$$

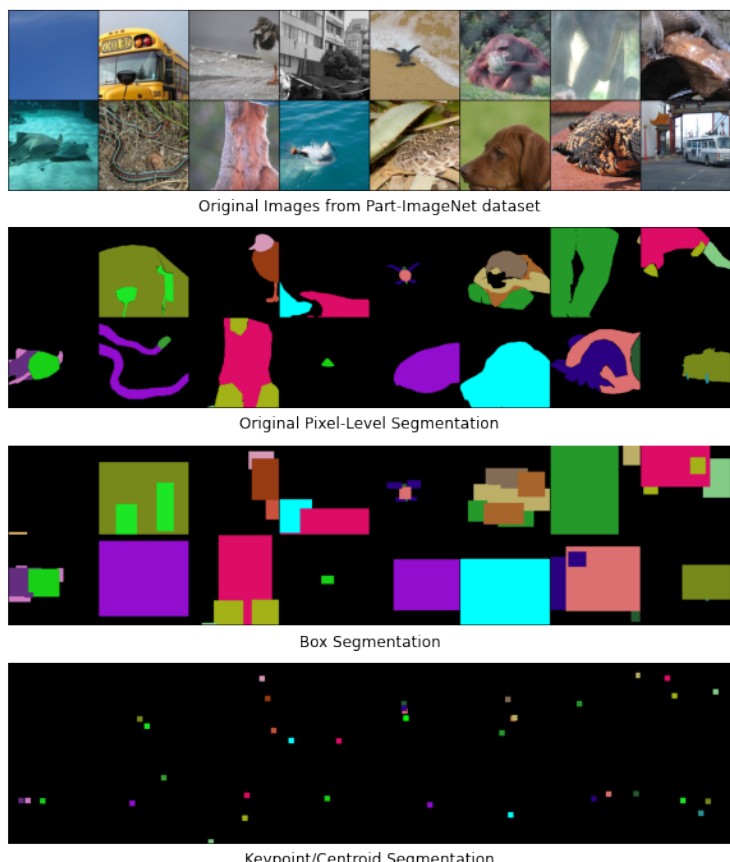

Figure 10: Random examples of part bounding-box labels and centroid labels used in the experiment in Section 5.4.

Table 14: Clean and adversarial accuracy of the downsampled part models trained with *object-level segmentation labels instead of part-level*. The model is adversarially trained (PGD) on Part-ImageNet with different values of $c_{\text{seg}}$. The adversarial accuracy is computed by AutoAttack and PGD attack.

| Models | $c_{\text{seg}}$ | Clean Accuracy | AutoAttack Accuracy | PGD Accuracy |
|---|---|---|---|---|
| Downsampled Part Model (Best) | - | 85.6 | 39.4 | 45.4 |
| Downsampled Part Model w/ Object Segmentation | 0.1 | **83.5** | 39.2 | 45.4 |
| | 0.3 | 81.3 | 37.9 | 44.2 |
| | 0.5 | 82.8 | **39.3** | **45.5** |
| | 0.7 | 81.6 | 38.0 | 45.1 |
| | 0.9 | 82.0 | 37.9 | 44.9 |

The first term is the mean squared error of the predicted centroids and the ground truth. The second ensures that the segmenter predicts masks of the correct class. For this, we use the cross-entropy loss with the logits being the sum of pixel-wise predictions across all parts of each object class.

### D.5    PART SEGMENTATION VS OBJECT SEGMENTATION

We conduct an ablation study to test whether the part-level annotation is necessary to improve the adversarial training. Can it be substituted with an object-level annotation which is cheaper to label? To answer this question, we train downsampled "part" models using *the object-segmentation labels* instead of the part-level annotation. Table 14 clearly indicates that the models trained on the object-level annotation achieve lower clean and adversarial accuracy compared to ones trained on the

Table 15: A simple semi-supervised technique (pseudo-labeling) can almost completely replace the full supervision needed for the part segmentation labels.

| Models | Num. Train Samples | Num. Seg. Labels | Clean Acc. | Adv. Acc. |
|---|---|---|---|---|
| ResNet-50 (baseline) | 20K | None | 74.7 | 37.7 |
| Downsampled part model | 20K | 2K (GT) | 78.7 | 38.9 |
| | 20K | 2K (GT) + 18K (pseudo) | 84.9 | 39.8 |
| | 20K | 20K (GT) | 85.6 | 39.4 |
| ResNet-50 (baseline) | 40K | None | 77.7 | 41.9 |
| Downsampled part model | 40K | 2K (GT) + 38K (pseudo) | 87.1 | 44.5 |

Table 16: Accuracy on the three generalized robustness benchmarks comparing the Downsampled part models with and without the background channel.

| Models | Common Corruptions | Background-vs-Foreground | Shape-vs-Texture |
|---|---|---|---|
| ResNet-50 | $82.3 \pm 1.6$ | $58.6 \pm 4.2$ | $40.6 \pm 1.8$ |
| Downsampled Part Models (w/ Background) | $85.5 \pm 0.8$ | $65.1 \pm 0.8$ | $44.7 \pm 2.6$ |
| Downsampled Part Models (w/o Background) | $85.5 \pm 1.8$ | $64.2 \pm 2.2$ | $45.1 \pm 2.3$ |

part-level annotation. This experiment suggests that training with object segmentation does improve adversarial training compared to the baseline, but using part segmentation can achieve even better results. Intuitively, the part annotation is more fine-grained and contains more information than the object one. So it is likely that stronger learning signals lead to higher robustness.

## D.6 EXTENDED EXPERIMENTS ON TRAINING WITH FEWER PART SEGMENTATION LABELS

In this section, we attempt to further reduce the labeling costs, using semi-supervised learning. **We show that using only 10% of all the segmentation labels (~2K samples) yields a model almost as good as the one using all the labels.** Specifically, we first train a part segmentation model on those 10% of images (~2K images or 175 per class) and use that model to generate pseudo-labels (predicted segmentation masks) on the remaining 90% of images. Then, we combine these pseudo-labels and the ground-truth labels to train a new part model. As shown in Table 15, this model performs about as well as the one trained with segmentation labels for 100% of training images (3rd vs 4th row or 84.9%/39.8% clean/robust accuracy vs 85.6%/39.4%) and performs significantly better than a model trained with no segmentation labels (3rd vs 1st row or 84.9%/39.8% vs 74.7%/37.7%).

Next, to test scaling, we double the training set size (from 20K to 40K) of PartImageNet by drawing additional samples from ImageNet, with class labels but no additional segmentation labels. The two bottom rows of Table 15 compares our part model to the baseline where a model is trained with this extra data but no segmentation labels. It shows that our part model scales well with more training data: it benefits from extra training data similarly to the normal model and still outperforms it by a large margin (10% clean and 3% adversarial accuracy). Here, the effective number of part segmentation labels is only 5% of all training samples (2K of 40K).

## D.7 EFFECTS OF BACKGROUND REMOVAL

We repeat the same experiments, measuring both adversarial and generalized robustness, on the Downsampled part models that remove the background. Specifically, we drop the background channel of the predicted segmentation mask by the segmenter before passing it to the second-stage classifier.

In summary, our results show that whether the predicted background channel is included or not has little effect on the accuracy. The model without background has 0.8% lower clean accuracy and the same adversarial accuracy as the one with the background channel. The result on the generalized robustness benchmarks in Table 16 also portrays a similar story: the Downsampled part models with and without the background perform similarly (within margins of error) but are still clearly better

Table 17: Clean and adversarial accuracy of the downsampled part models trained on Part-ImageNet with different values of downsampling output sizes. All of the models here are trained with a learning rate of 0.1, weight decay of $5 \times 10^{-4}$, and $c_{\text{seg}}$ of 0.5. The adversarial accuracy is computed by AutoAttack and PGD attacks.

| Downsampling Output Sizes | Clean Accuracy | AutoAttack Accuracy | PGD Accuracy |
|---|---|---|---|
| $1 \times 1$ | 83.9 | 39.9 | **45.9** |
| $2 \times 2$ | 84.0 | 39.4 | 45.5 |
| $4 \times 4$ | 83.9 | 39.9 | 45.3 |
| $8 \times 8$ | 83.0 | 39.5 | **45.9** |
| $32 \times 32$ | 83.0 | 38.7 | 45.4 |
| $128 \times 128$ | **84.3** | **40.0** | 45.7 |

Table 18: Clean and adversarial accuracy of the part model variants trained on Part-ImageNet with different backbone architectures.

| Backbone Arch. | Models | Clean Acc. | Adv. Acc. |
|---|---|---|---|
| EfficientNet B4 | Baseline | 83.1 | 37.1 |
| | Part Model | 88.4 | 41.4 |
| ResNeXt-50 32x4d | Baseline | 77.4 | 36.9 |
| | Part Model | 86.4 | 39.6 |

than ResNet-50. This experiment suggests that the second-stage classifier can learn to ignore the background pixels automatically. So there is no clear benefit to dropping them.

## D.8    EFFECTS OF THE DOWNSAMPLED SIZE

Table 17 shows the performance of the downsampled part model when the output size of the pooling layer changes. Across all the sizes from 1 to 128, both the clean and the adversarial accuracy barely change; the gap between the largest and the smallest numbers is under 1.3 percentage points. This suggests that the performance of the downsampled part model is insensitive to the choice of the downsampling output size. We use the downsampling size of $4 \times 4$ throughout this paper, but almost any other number can be used since the difference is not significant.

### D.8.1    EFFECTS OF THE BACKBONE ARCHITECTURE

We train both the baseline and our part models with two additional backbone architectures with a similar size to ResNet-50 (EfficientNet-B4 and ResNeXt-50-32x4d). We find that **our part model consistently improves over the baseline across all architectures (5-9% increase in clean and 3-4% in adversarial accuracy).**

## D.9    ADVERSARIAL ROBUSTNESS RESULTS ON THE REMAINING PART MODELS

In this section, we include the robustness results on the other two part-based models we omit from the main text, i.e., the two-headed and the pixel part models. We report the accuracy of the models trained with five different values of $c_{\text{seg}}$ for completeness and for displaying a minor trade-off between the clean and the adversarial accuracy. However, comparing the best models alone would be sufficient.

Table 19 suggests that the two-headed part models perform similarly to the downsampled variant and slightly worse than the bounding-box one when all of them are adversarially trained with PGD. On the other hand, the pixel part models have consistently lower accuracy than the other part models by roughly 1–2 percentage points. This result confirms our hypothesis on the importance of the spatial information as mentioned in Section 3.1 as well as Appendix C.2.

Table 19: Clean and adversarial accuracy of the part model variants adversarially trained (PGD) on Part-ImageNet with different values of $c_{\text{seg}}$. The adversarial accuracy is computed by AutoAttack and PGD attack ($\epsilon = 8/255$). For comparison, we add the first two rows for the two best part models we reported in the main paper. The highest accuracy in each column of each model is bold.

| Models | $c_{\text{seg}}$ | Clean Accuracy | AutoAttack Accuracy | PGD Accuracy |
|---|---|---|---|---|
| Downsampled Part Model (Best) | - | 85.6 | 39.4 | 45.4 |
| Bounding-Box Part Model (Best) | - | 86.5 | 39.2 | 45.7 |
| Two-Headed Part Model | 0.1 | **86.1** | 38.9 | 44.7 |
| | 0.3 | 84.6 | 38.2 | 44.5 |
| | 0.5 | 85.4 | 39.2 | 44.6 |
| | 0.7 | 84.6 | 38.9 | 44.7 |
| | 0.9 | 85.7 | **39.4** | **44.9** |
| Pixel Part Model | 0.1 | **84.5** | 39.6 | 45.4 |
| | 0.3 | 83.0 | 38.5 | 45.1 |
| | 0.5 | 83.1 | 37.8 | 45.0 |
| | 0.7 | 83.3 | **39.7** | **46.0** |
| | 0.9 | 84.3 | 39.6 | 45.5 |

Table 20: Clean and adversarial accuracy of *the downsampled part models with concatenated input images* (see Appendix D.10). The model is adversarially trained (PGD) with different values of $c_{\text{seg}}$ on Part-ImageNet. The adversarial accuracy is computed by AutoAttack and PGD attack ($\epsilon = 8/255$).

| Models | $c_{\text{seg}}$ | Clean Accuracy | AutoAttack Accuracy | PGD Accuracy |
|---|---|---|---|---|
| Downsampled Part Model (Best) | - | 85.6 | 39.4 | 45.4 |
| Downsampled Part Model w/ Concat. Input | 0.1 | 82.2 | 37.7 | 44.4 |
| | 0.3 | **82.6** | 38.7 | 45.0 |
| | 0.5 | 79.9 | 38.7 | 44.8 |
| | 0.7 | 76.9 | 39.1 | **45.3** |
| | 0.9 | 72.7 | **39.5** | 44.1 |

## D.10 FEEDING INPUT IMAGES TO THE PART MODEL

In Section 3.1, we suggest that the classifier stage of the part models should not see the input image directly. We hypothesize that doing so opens up an opportunity for the attacker to bypass the more robust segmenter and influence the small and less robust classifier. This essentially defeats the purpose of the segmentation and the part model overall. However, there is also a counterargument to this hypothesis. In theory, if the model is fed with both the image and the predicted segmentation mask, it strictly receives more information. When adversarially trained, the model can then learn to ignore the image if it is deemed non-robust. Hence, this model should be strictly better or at least the same as the one that sees only the segmentation mask.

To find out which hypothesis holds, we create a variant of the downsampled part model by concatenating the input image to the predicted segmentation mask before being fed to the classifier stage. We then compare this model to the original downsampled part model. The empirical results support our hypothesis. Table 20 shows that this input-concatenated downsampled part model performs slightly worse compared to the original version. We leave it to future work to unveil the underlying reasons that make the model less robust when more information is presented to it.

## D.11 DETAILED RESULTS ON THE GENERALIZED ROBUSTNESS

We also evaluate adversarially-trained models on the generalized robustness datasets, in addition to the normally trained ones reported in Section 5.1. Fig. 11 shows *the robust accuracy*[6] on the three benchmarks with respect to the clean accuracy of the models. The number next to each data point

---

[6] In this section, we will refer to the accuracy on the generalized robustness benchmarks as *the robust accuracy*. On the other hand, *the adversarial accuracy* still denotes the accuracy under adversarial attacks.

Table 21: Comparisons of the models on their generalized robustness. Higher is better. For each of the model types, we report two models, (A) and (B), trained with a different set of hyperparameters. Model (A) is the one with the highest accuracy on the shape-vs-texture benchmark, and model (B) is the one with the highest accuracy on both the spurious correlation and the common corruption benchmarks. All models are trained on Part-ImageNet without adversarial training.

| Models | Shape-vs-Texture Bias | Spurious Correlation | Corruption Robustness |
|---|---|---|---|
| ResNet-50 (A) | **40.6 ± 1.8** | 57.7 ± 2.0 | 81.5 ± 1.2 |
| ResNet-50 (B) | 40.5 ± 1.9 | **58.6 ± 4.2** | **82.3 ± 1.6** |
| Downsampled Part Model (A) | **44.7 ± 2.6** | 62.9 ± 2.1 | 84.3 ± 0.4 |
| Downsampled Part Model (B) | 43.4 ± 2.1 | **65.1 ± 0.8** | **85.5 ± 0.8** |
| Bounding-Box Part Model (A) | **45.7 ± 2.7** | 60.0 ± 2.0 | 82.0 ± 1.1 |
| Bounding-Box Part Model (B) | 44.4 ± 3.2 | **65.1 ± 2.1** | **85.8 ± 0.7** |

Table 22: Accuracy for each corruption type from the common corruption benchmark, averaged across 10 random seeds during training. The highest number on each row is bold.

| Corruption Type | ResNet-50 | Downsampled Part Model | Bounding Box Part Model |
|---|---|---|---|
| Gaussian Noise | 82.3 | 84.3 | **84.7** |
| Shot Noise | 82.4 | 84.1 | **84.5** |
| Impluse Noise | 80.8 | 83.6 | **84.2** |
| Defocus Blur | 81.7 | 86.1 | **86.3** |
| Glass Blur | 80.3 | **84.0** | 83.5 |
| Motion Blur | 79.1 | **83.5** | 83.5 |
| Zoom Blur | 67.4 | 70.1 | **70.8** |
| Snow | 75.1 | 80.1 | **80.7** |
| Frost | 78.8 | 83.4 | **83.6** |
| Fog | 86.7 | 90.5 | **90.9** |
| Brightness | 94.4 | 96.2 | **96.4** |
| Contrast | 71.0 | 74.5 | **75.2** |
| Elastic Transform | 88.2 | 92.3 | **92.4** |
| Pixelate | 92.6 | 94.6 | **94.8** |
| JPEG Compression | 93.4 | 95.1 | **95.2** |

represents the adversarial accuracy, and due to the (adversarial) robustness-accuracy trade-off, the points on the top right corner generally have higher clean accuracy but lower adversarial accuracy.

It is clear that there is a strong correlation between the clean accuracy and the robust accuracy on all three benchmarks. A similar trend is also observed in Taori et al. (2020). In contrast, we do not find that adversarial training improves the common corruption robustness, spurious correlation robustness, or shape bias. Nonetheless, we emphasize that **the part models still outperform the ResNet-50 at almost all levels of clean accuracy across all types of robustness studied.**

Table 21 depicts the full results of the generalized robustness evaluations on the part-based models and the baseline. As mentioned in Section 5.1, we conduct a hyperparameter search in order to find the best model for each of the benchmarks we test on. In this section, we report the robust accuracy of these models on all the benchmarks, not only the one that they perform the best in. Generally, we would have three rows per model architecture, one per dataset. However, interestingly, the best-performing models on the spurious correlation benchmark and the best-performing models on the corruption robustness benchmark are coincidentally the same models, i.e., model (B) in Table 21. On the other hand, the models (A) are only the best in the shape-vs-texture bias benchmark. This trend is consistent on the ResNet-50 as well as our part models. This phenomenon could be some sort of trade-off behavior. However, more experiments are needed to make further conclusions.

Table 22 shows a breakdown of the corruption robustness accuracy for each corruption type. This result confirms that the two part-based models outperform the ResNet-50 baseline on **all corruption types**, not only the mean. The bounding-box part model also achieves very slightly higher robust accuracy than the downsampled one across most of the corruption types.

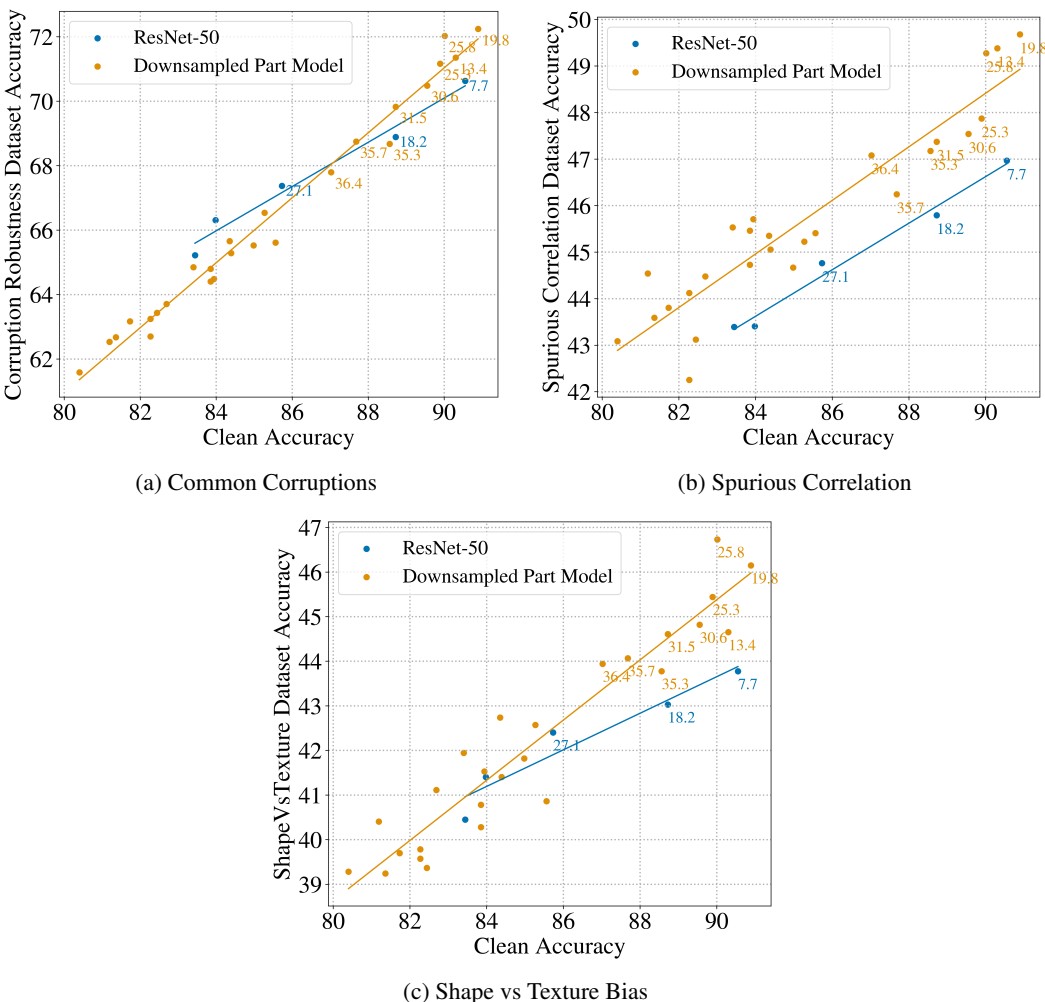

(a) Common Corruptions

(b) Spurious Correlation

(c) Shape vs Texture Bias

Figure 11: Plots of the robust accuracy on each of the three generalized robustness benchmark with respect to the clean accuracy. Each data point represents one adversarially trained model. The number next to each point is the adversarial accuracy (AutoAttack, $\epsilon = 8/255$). Generally, in the region where the clean accuracy is high, the part-based models outperform the ResNet-50 baseline on all accuracy metrics.

## D.12 ADDITIONAL VISUALIZATION OF THE PART MODELS

We provide additional visualization of the outputs of our part-based models on all three datasets. Fig. 12 shows a similar visualization to Fig. 5 but for the downsampled part model. The same visualization for Cityscapes (resp. PASCAL-Part) on the downsampled and the bounding-box part models can be found in Fig. 13 (resp. Fig. 14).

Apart from the ones trained on PASCAL-Part, our part-based models segment the object part fairly well even though some amount of detail and small parts are sometimes missed. In most of the misclassified samples, the predicted segmentation masks are also incorrect. This is particularly true for the PGD adversarial images. This observation qualitatively confirms that the classifier stage of the part model depends on and agrees with the segmentation mask as expected.

We suspect that the poor prediction of the segmentation on the PASCAL-Part dataset may be attributed to the small number of training samples. PASCAL-Part has about one order of magnitude fewer training samples compared to the other two datasets. Nevertheless, the segmentation labels still prove to be very helpful in improving the adversarial training, potentially also due to the fact that the

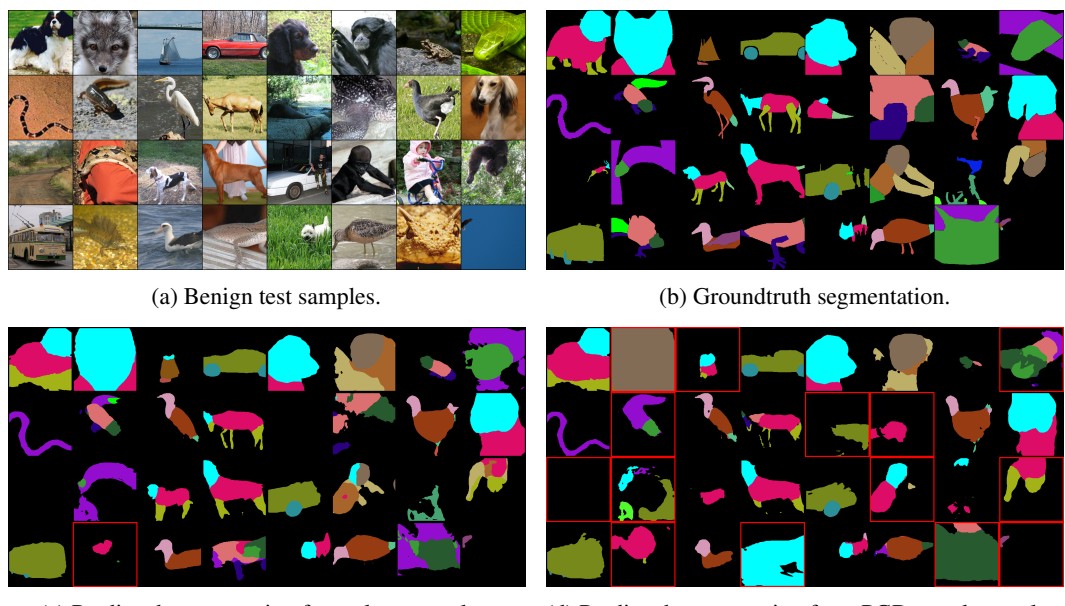

(a) Benign test samples.

(b) Groundtruth segmentation.

(c) Predicted segmentation from clean samples.

(d) Predicted segmentation from PGD-attack samples.

Figure 12: Visualization of *the downsampled part model on Part-ImageNet*: (a) randomly selected clean test samples, (b) the corresponding groundtruth segmentation mask, (c) their predicted segmentation mask from the segmenter, and (d) the predicted segmentation mask when the samples are perturbed by PGD attack ($\epsilon = 8/255$). Segmentation masks corresponding to misclassified samples are indicated by a red box.

number of training data is small. One interesting future direction is to study the relationship between the numbers of class labels and segmentation labels with respect to robustness.

# E   SOCIETAL IMPACT

Our work focuses on improving the adversarial robustness of neural networks with the goal of creating secure and reliable models. We strictly propose a new "defense" technique and do not contribute to any attack algorithm. We believe that our work will benefit not only the research community but also machine learning practitioners and eventually, society overall. We hope that our work will be extended to improve the effectiveness of adversarial training in practice, leading to broader adoption of deep learning as well as preventing potential vulnerabilities and failures in the future.

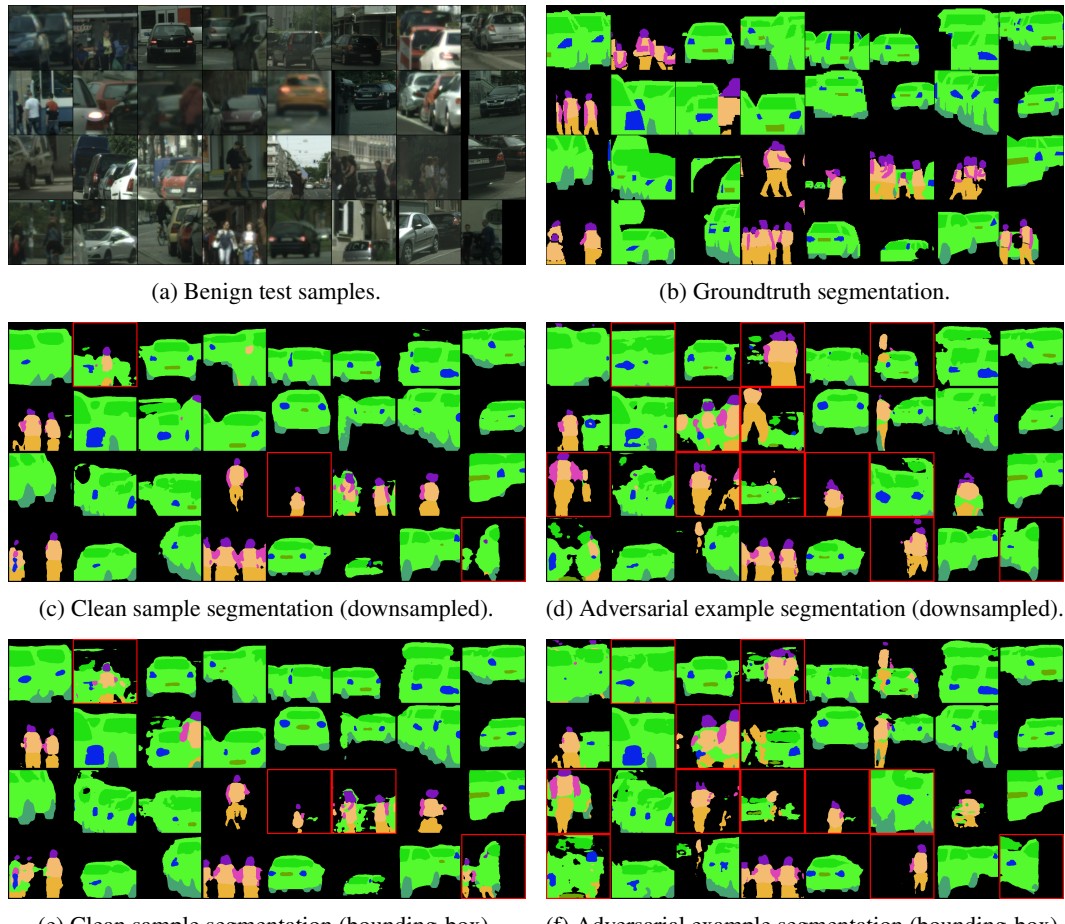

(a) Benign test samples.

(b) Groundtruth segmentation.

(c) Clean sample segmentation (downsampled).

(d) Adversarial example segmentation (downsampled).

(e) Clean sample segmentation (bounding-box).

(f) Adversarial example segmentation (bounding-box).

Figure 13: Visualization of the part model trained on *Cityscapes*: (a) randomly selected clean test samples, (b) the corresponding groundtruth segmentation mask. (c) and (d) are the predicted segmentation mask from the downsampled part model on clean and adversarial samples (PGD attack with $\epsilon = 8/255$), respectively. (e) and (f) are the segmentation masks from the bounding-box model. Segmentation masks corresponding to misclassified samples are indicated by a red box.

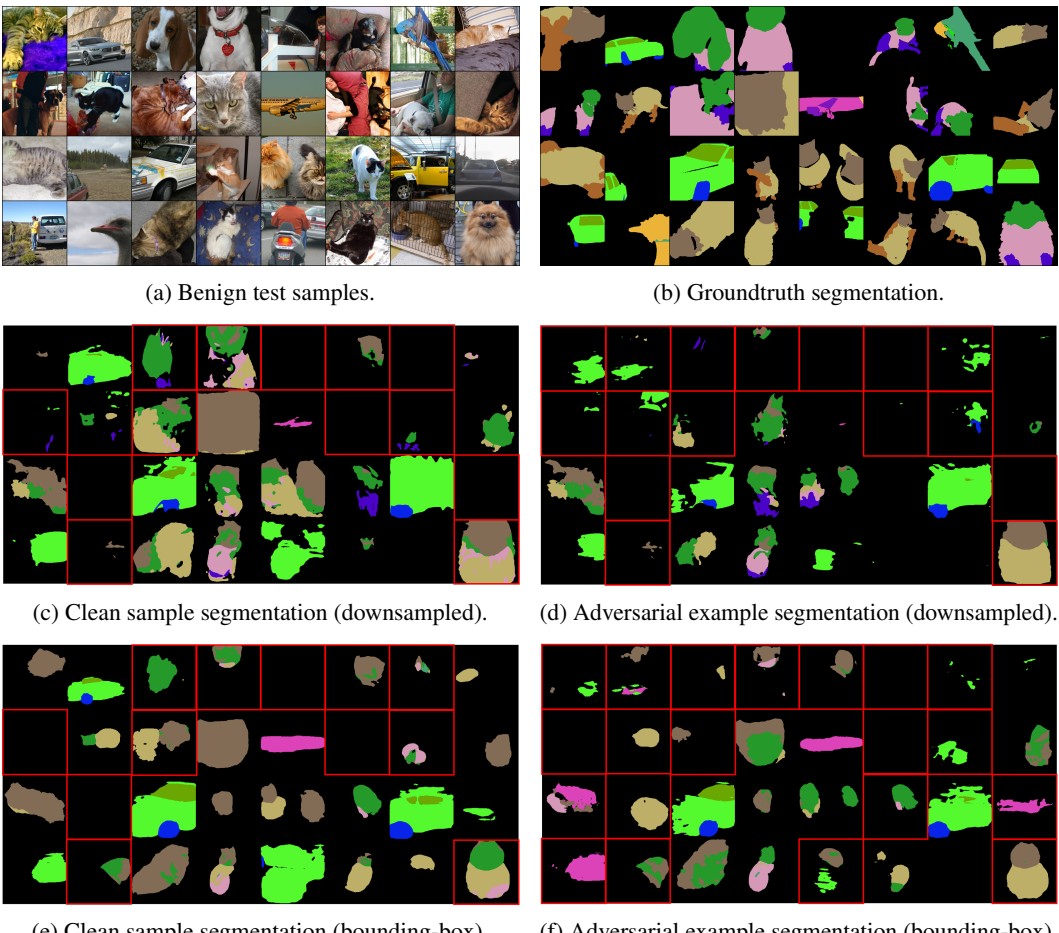

(a) Benign test samples.

(b) Groundtruth segmentation.

(c) Clean sample segmentation (downsampled).

(d) Adversarial example segmentation (downsampled).

(e) Clean sample segmentation (bounding-box).

(f) Adversarial example segmentation (bounding-box).

Figure 14: Visualization of the part model trained on *PASCAL-Part*: (a) randomly selected clean test samples, (b) the corresponding groundtruth segmentation mask. (c) and (d) are the predicted segmentation mask from the downsampled part model on clean and adversarial samples (PGD attack with $\epsilon = 8/255$), respectively. (e) and (f) are the segmentation masks from the bounding-box model. Segmentation masks corresponding to misclassified samples are indicated by a red box.

