# OpenReview forum: "Part-Based Models Improve Adversarial Robustness"
_ICLR.cc/2023/Conference — ICLR 2023 poster_

### Official Review · Reviewer_8aoz · 2022-10-24

**Confidence:** 4
**Clarity, Quality, Novelty And Reproducibility:** The clarity, quality, and reproducibi…
**Correctness:** 4
**Technical Novelty And Significance:** 3
**Empirical Novelty And Significance:** 3
**Recommendation:** 6

**Strength And Weaknesses:**

Strengths:
+ This paper proposes an interesting framework to use the segmentation results to help the classification robustness.
+ The experiments are sufficient, and the results demonstrate that the proposed method is effective compared with baselines.
+ This paper is technically sound and well-written.

Weakness:
+ This segmentation-based classification framework needs the introduction of segmentation labels, greatly limiting its application scenarios.
+ This paper only considers a ResNet baseline. It’s hard to identify its effectiveness across different model architectures.
+ Because the proposed method is based on a segmentation framework, the segmentation results will affect the classification. I wonder whether segmentation parts’ training will affect the training time and convergence speed.


**Summary Of The Paper:**

This paper proposes a segmentation-based robust classification framework, combining a part segmentation model with a tiny classifier, and achieves a good performance. Furthermore, this paper offers two part-based models, a down-sampled part-based model and a bounding-box part-based model. Both achieve both higher accuracy and higher adversarial robustness than a ResNet-50 baseline.

**Summary Of The Review:**

This paper proposes a segmentation-based robust classification framework, combining a part segmentation model with a tiny classifier, and achieves a good performance. But my major concern is that its application scenario is limited because of the need for segmentation labels.

---

> ### Author Response · Authors · 2022-11-19
> **Response to Review 8aoz**
>
> We thank the reviewer for the thoughtful questions and constructive comments. Please find our review-specific responses below and common responses in the main post above [the main post above](https://openreview.net/forum?id=bAMTaeqluh4&noteId=z6SqdgBC8X5).
>
> #### Q: Requirement on the extra segmentation labels.
> (Please see the common response above)
>
> #### Q: Performance with other backbone models.
> (Please see the common response above)
>
> #### Q: Does part segmenter’s training affect the training time and convergence speed?
>
> To answer this question, we train our part models for 20, 30, 40, 50 (default), and 100 epochs. We found that the normal and the part model seem to converge to approximately the best performance between 40 and 50 epochs. It does not appear that the part model converges more slowly than the normal one in terms of the number of epochs.
>
> Table F: Models’ performance with different numbers of training epochs.
> | Train Epochs | Models                 | Clean Acc. | Adv. Acc. |
> |--------------|------------------------|-----------:|----------:|
> | 20           | Resnet-50 (baseline)   |       68.5 |      34.9 |
> | 20           | Downsampled part model |       80.4 |      37.9 |
> | 30           | Resnet-50 (baseline)   |       71.9 |      36.4 |
> | 30           | Downsampled part model |       83.2 |      38.5 |
> | 40           | Resnet-50 (baseline)   |       74.5 |      37.0 |
> | 40           | Downsampled part model |       83.4 |      39.2 |
> | 50           | Resnet-50 (baseline)   |       74.7 |      37.7 |
> | 50           | Downsampled part model |       85.6 |      39.4 |
> | 100          | Resnet-50 (baseline)   |       76.3 |      36.6 |
> | 100          | Downsampled part model |       84.3 |      40.1 |

---

### Official Review · Reviewer_moqQ · 2022-10-25

**Confidence:** 4
**Correctness:** 3
**Technical Novelty And Significance:** 3
**Empirical Novelty And Significance:** 2
**Recommendation:** 6

**Clarity, Quality, Novelty And Reproducibility:**

Very clear paper. It was easy to follow, easy to understand the core idea of the paper, and easy to grab the superiority of the paper. Very well written.
The experimental validation is good enough though not throughout.
I think the work is novel, but I have some concerns as summarized above. Please see my weakness section.
The code was included in the supplementary material, though I did not check it. So, it should be reproducible.


**Strength And Weaknesses:**

[Strength]
I think the part-based model is a new idea and interesting.
It was interesting to me to see that even the clean dataset accuracy can be improved by the proposed method. In conventional approaches, it was common that the robustness to attacks can be enhanced by sacrificing the clean dataset accuracy.

[Weaknesses]
I have some concerns in the proposed method.

I am afraid the comparison to the prior work is not well demonstrated. The authors should compare their method with previous methods that can enhance the robustness to the attacks.

I am afraid that the practicability of the proposed model is quite limited because the labels for the semantic segmentation are required. I understand a smaller number of labels will still be valid as shown in the experimental validation, the requirement to have semantic segmentation labels would severely limit the applications of the proposed model.

The robustness to the adversarial attacks that attack semantic segmentation models is not discussed. Once this method is known to the attackers, it is apparent the attackers would attack the semantic segmentation model first. So, the authors may want to discuss its robustness.



**Summary Of The Paper:**

This paper presents two stage end-to-end image recognition model that is robust against adversarial attacks. The model consists of the semantic segmentation module that tries to divide the objects of interest in the scene into several sub-regions and a part-based recognition model that makes the final outputs. Experimental results show that the proposed model can make the model very robust to the adversarial attacks. Detailed discussion on its performance is also given.

**Summary Of The Review:**

I admit that the paper is interesting and well-written. However, there are pros and cons in this paper. By considering the trade-off between them, I am a bit on the rejection side of the paper. Please see my weakness section.

---

> ### Author Response · Authors · 2022-11-19
> **Response to Review moqQ**
>
> We thank the reviewer for the thoughtful questions and constructive comments. Please find our review-specific responses below and common responses in the main post above [the main post above](https://openreview.net/forum?id=bAMTaeqluh4&noteId=z6SqdgBC8X5).
>
> #### Q: Requirement on the extra segmentation labels.
> (Please see the common response above)
>
> #### Q: Attacking the segmenter directly does not work well.
>
> Thank you for sharing the concern that the robustness of the segmentation model was not discussed. Attacking the segmenter is a natural step that we have already taken to test the robustness of our model. In fact, we experimented with the best possible adaptive attacks on the segmenter that we could come up with. This is mentioned in the last paragraph of page 7 (Section 4) and explained in detail in Appendix D.3. To summarize, we have tried (i) a single-stage attack that considers both segmentation and classification losses at the same time and (ii) four variants of two-stage attacks that first focus on attacking the segmenter then build on top of it to attack the classification task. **None of these attacks works as well as using AutoAttack on the part model end-to-end**, which we used to report the robustness numbers throughout the paper. Please refer to Table 12 for the full comparison. We also further investigate why the attack on the segmenter does not work well. Please see the response to Review WbK2 for more detail.
>
> Additionally, for the rebuttal, we tried one more adaptive attack that tries to break the part model in the inverse order (we will call it the "inverse attack"). First, it attacks the second-stage classifier by coming up with the worst-case mask, and then it tries to fool the segmenter into producing this mask. **Nonetheless, this attack is also less effective than the default PGD attack that targets the end-to-end model**, achieving adversarial accuracy of 73.9% (vs 45.4%) and 77.1 (vs 45.7%) for the Downsampled and the Bounding-box part models, respectively.
>
> #### Q: Comparison to the prior work is not well demonstrated.
>
> We are not entirely sure which prior work is referred to here. Please feel free to clarify. We do compare our work to adversarial training, which is the state-of-the-art algorithm for adversarial robustness in this setting. There has been considerable work on fine-tuning adversarial training, e.g., with extra data or augmentation, and our approach is orthogonal to and can be combined with that work.

---

> > ### Comment · Reviewer_moqQ · 2022-11-27
> > **After reading the authors' responses**
> >
> > I am satisfied with the authors responses and would like to change to the positive side.

---

### Official Review · Reviewer_cNQX · 2022-10-25

**Confidence:** 3
**Correctness:** 4
**Technical Novelty And Significance:** 2
**Empirical Novelty And Significance:** 2
**Recommendation:** 6

**Clarity, Quality, Novelty And Reproducibility:**

The clarity is good; so does the quality. Regarding the novelty, the reviewer feels the proposed method/direction is probably not very novel, people may have already exploited similar ideas in 3D data, where the parts of 3D objects are more salient than the "parts" of objects in 2D images. Reproducibility is guaranteed as the codes provided in the appendix.

**Strength And Weaknesses:**

The studied direction is important, and this paper takes a step further on deep learning robustness with the help of fine-grained annotations. Overall the use of previous techniques and the design principles (part-based, disentangling irrelevant features, and location-aware) are reasonable. Their approach is properly motivated, and the paper is easy to follow.

Experiments are conducted on three datasets, Part-ImageNet, Cityscapes, and PASCAL-Part, and shown the proposed part-based model achieved consistent improvements under both vanilla training and adversarial training, while the improvements further with adversarial training are limited. The experiments of attacking the segmenter model strongly prove the robustness of the proposed method. Table 6 shows the proposed method can also utilize BB and Centroids to improve model robustness.

One of the major limitations of the proposed method is the use of extra data annotations (part mask/BB). Also, the part of objects in 2D images is not well defined. It is unknown if Vision Community has a generable "object part segmentor" for general 2d images. It is ok to study this in the existing dataset for academic purposes, while the reviewer is unsure if the proposed method can scale up to real-world questions.

According to 3.5.2, the author has similar #params and training epochs for the baseline and the part-based model. It is good for fair comparisons, but, the reviewer wonders if the part-based model can achieve higher performance with more training epochs? By using more fine-grained information, it is expected to train longer somehow.

For the current manuscript, the proposed method is only examined in ResNet-50. The reviewer is curious if the proposed method can work on other ConvNets and ViTs? The use of part masks is natural to introduce Vision-Transformers, which is one of the popular backbones for vision tasks nowadays. It would be good to see the proposed method can model-agnostically improve the robustness.

**Summary Of The Paper:**

This paper studied how to further improve deep learning model robustness with fine-grained annotations (object parts in 2D images). They proposed a part-based classification model and compared it with a vanilla ConvNet-based classifier (ResNet-50). On three common benchmarks (Part-ImageNet, Cityscapes, and PASCAL-Part), their experiments show the proposed part-based method consistently outperforms the holistic counterpart. They also conducted experiments around attacking the segmentation model and utilizing part-(Segmentation, BB, Centroids) information to further prove the effectiveness of the proposed method.

**Summary Of The Review:**

Overall, the reviewer believes the empirical studies included in this paper can help the related community. However, the requirement of image-part (ill-defined) masks may significantly limit the practical use of the proposed approach.

---

> ### Author Response · Authors · 2022-11-19
> **Response to Review cNQX**
>
> We thank the reviewer for the thoughtful questions and constructive comments. Please find our review-specific responses below and common responses in the main post above [the main post above](https://openreview.net/forum?id=bAMTaeqluh4&noteId=z6SqdgBC8X5).
>
> #### Q: Requirement on the extra segmentation labels.
> (Please see the common response above)
>
> #### Q: Performance with other backbone models.
> (Please see the common response above) We do not have the resources to experiment with ViTs at this moment but plan to include them in the future.
>
> #### Q: The part-based model can achieve higher performance with more training epochs?
>
> To answer this question, we trained both the baseline and the part model for 100 epochs (twice as long as all the models reported in the paper) but do not observe a significant improvement (ResNet-50: 74.7% clean/37.7% robust to 76.3%/36.6%, downsampled part model: 85.6%/39.4% to 84.3%/40.1%).
>
> #### Q: "The part of objects in 2D images is not well defined. It is unknown if Vision Community has a generable "object part segmentor" for general 2d images."
>
> We agree that the notion of "parts" is not a completely well-defined concept, even among humans. The definition of the parts we use is arguably reasonable for humans, but might not be optimal for robustness. We suspect that any reasonable definition of parts that results in consistent labeling would suffice for our part models. We leave it to future work to study whether one can come up with an even better definition of parts. An adaptive or learnable method for coming up with parts could be a promising future direction.

---

### Official Review · Reviewer_WbK2 · 2022-10-28

**Confidence:** 3
**Correctness:** 2
**Technical Novelty And Significance:** 2
**Empirical Novelty And Significance:** 2
**Recommendation:** 5

**Clarity, Quality, Novelty And Reproducibility:**

* Clarity: most part of the paper is clear enough
* Quality: see details in the "Weaknesses" above
* Novelty: good
* Reproducibility: should be good

**Strength And Weaknesses:**

### Strength
* The investigated problem is very important and difficult, to defend against white box adversarial attack.
* The model architecture is somewhat straightforward and can be adapted to many other segmentation models.

### Weaknesses
* The conclusion in Table 1 is a bit confusing: the PGD adversarial training is more robust under adversarial attack, but the proposed method does not improve the robustness a lot with PGD adversarial training (e.g. 37.8 -> 38.5 on the Pascal-Part dataset). Considering the results reported in Table 5, training without segmentation supervision even leads to better adversarial robustness. So that means the extra supervision is more helpful towards clean accuracy but not on adversarial accuracy?
* The proposed method need extra annotation of segmentation masks, which may not readily available and existing datasets with segmentation masks may not fit the classification task without non-trivial modification. This limits the scopes the proposed approach can be applied on.
* It's a bit surprising that in appendix D.3, directly attacking the segmentation module does not work well. Can the authors give a bit more explanation about the reason and the detailed analysis of the segmentation attack results (in addition to the classification accuracy). As in [a], it seems that the segmentation task can be easily attacked.

[a] Adversarial Examples for Semantic Segmentation and Object Detection

**Summary Of The Paper:**

This paper aims at improving adversarial robustness especially for the classification task, under the white box attack settings. Authors argue that richer annotations such as part segmentation can be helpful to improve the robustness. To this end, authors crafted several classification datasets with part segmentation masks available, and proposed a semantic segmentation based classification framework. Experiments with AutoAttack demonstrate the effectiveness of the proposed framework.

**Summary Of The Review:**

This paper proposes to improve adversarial robustness, however it requires extra annotation and does not improve adversarial accuracy with PGD adversarial training.

---
Post rebuttal updates:
I thank authors' response, and it partially addresses my concerns and explains why part segmentation is more robust. However, the Table 1 still concerns me that the proposed method is not very effective on the strong PGD trained model, when defending adversarial attack (e.g. 37.8->38.5). Thus I keep my initial ratings unchanged.

---

> ### Author Response · Authors · 2022-11-19
> **Response to Review WbK2**
>
> We thank the reviewer for the thoughtful questions and constructive comments. Please find our review-specific responses below and common responses in [the main post above](https://openreview.net/forum?id=bAMTaeqluh4&noteId=z6SqdgBC8X5).
>
> #### Q: Requirement on the extra segmentation labels.
> (Please see the common response above)
>
> #### Q: Attacking the segmenter directly does not work well.
>
> Review WbK2 asks: why is this attack ineffective when Xie et al. [2017] have shown that it is possible to attack segmentation and detection models? The answer to this lies in the fact that **our segmenter has been adversarially trained** (end-to-end together with the classifier) whereas the models used in Xie et al. [2017] are only normally trained. To confirm this, we run PGD attacks on the segmenter part of our part models. Table C below shows that without adversarial training, it is easy to attack the part model and reduce the segmentation accuracy to under 10% (the right-most column). This is in line with Xie et al. [2017]. On the other hand, once adversarial training is used, we have a much more robust segmenter with over 60% adversarial accuracy. Since our segmenter is robust, the part model as a whole is also robust.
>
> Table C: Effectiveness of the segmenter-only attack on our part models.
> | Models                  | Adv. Train | Class Adv. Acc. | Seg. Adv. Acc. |
> |-------------------------|:----------:|----------------:|---------------:|
> | Downsampled part model  |      N     |            34.9 |            9.6 |
> | Downsampled part model  |      Y     |            60.9 |           62.6 |
> | Bounding-Box part model |      N     |            30.5 |            7.8 |
> | Bounding-Box part model |      Y     |            64.4 |           65.5 |
>
> The visualization of the predicted segmentation masks under this attack can be found [here](https://imgur.com/a/fsAXCCi). We can see that the normally trained model predicts almost entirely wrong masks (the whole image as one wrong part) whereas the adversarially trained model maintains mostly correct predictions.
>
> Table C shows that attacking the segmenter alone is less effective than attacking the part model end-to-end (60.9% vs 45.4% adversarial accuracy for the downsampled part model, 64.4% vs 45.7% for the bounding-box part model). This result agrees with our prior experiments on single-stage attacks: attacking the segmenter alone is equivalent to a single-stage attack with $c_{seg} = 1$, and we found that larger values of $c_{seg}$ consistently led to less effective attacks (see Table D right below or Appendix D.3, Table 11).
>
> Table D: Adversarial accuracy of the part model from PGD attack that focuses entirely on the classification loss, i.e., an end-to-end attack ($c_{seg}=0$), vs an attack on segmentation loss only ($c_{seg}=1$).
> | $c_{seg}$ |  0.0 |  0.1 |  0.3 |  0.5 |  0.7 |  0.9 |  1.0 |
> |-----------|-----:|-----:|-----:|-----:|-----:|-----:|-----:|
> | Adv. Acc. | 45.4 | 45.9 | 48.0 | 50.4 | 53.7 | 57.5 | 60.9 |
>
> - Xie et al., Adversarial Examples for Semantic Segmentation and Object Detection, ICCV 2017.
>
> #### Q: Is extra supervision helpful for clean accuracy but not adversarial accuracy?
>
> Yes, for PGD adversarial training. For TRADES, extra supervision improves the tradeoff between clean accuracy and adversarial accuracy: depending on the value of $\beta$ chosen, it is possible to use the extra supervision to increase clean accuracy or adversarial accuracy, or both.

---

### Author Response · Authors · 2022-11-19
**Common response to all reviews (1/2)**

We thank all reviewers for their thoughtful and constructive comments. We will be sure to incorporate the additional results here into the final version of the paper. Please find the review-specific responses in the comment under each review. Below are our responses to the common questions.

#### Q: Requirement on the extra segmentation labels (WbK2, cNQX, moqQ, 8aoz).

We certainly agree that the need for part segmentation labels comes at an increased cost, which has been acknowledged throughout the paper. Our primary goal in this paper is to demonstrate that it is possible to achieve **significant improvements in robust accuracy** using additional supervision. This result is particularly important since progress in the field has somewhat stagnated, and recent improvements through more training samples show diminishing returns [Gowal et al., 2021]. It is an open question whether the most cost-effective way to gain robustness is with more training samples or with richer supervision; our paper provides evidence for the first time that richer (segmentation-like) labels might be a cost-effective route to stronger robustness.  We hope our findings will stimulate follow-on research that explores how to achieve these benefits as cheaply as possible.

That said, we have tried a few approaches to reduce the labeling cost. First, the paper describes several experiments in reducing the labeling cost: using fewer segmentation labels (Section 5.3), using cheaper forms of annotation (Section 5.4), and using object segmentation instead of part segmentation (Appendix D.5). One promising direction is to use bounding box labels, which are much cheaper to collect, instead of pixel-wise segmentation labels. This achieves the same level of robustness as pixel-wise segmentation labels, with clean accuracy only 1.5% lower (Section 5.4), and much lower labeling cost.

Second, in response to reviews, we have conducted new experiments to further reduce the labeling costs, using semi-supervised learning. **We show that using only 10% of all the segmentation labels (\~2K samples) yields a model almost as good as the one using all the labels.** Specifically, we first train a part segmentation model on those 10% of images (\~2K images or 175 per class) and use that model to generate pseudo-labels (predicted segmentation masks) on the remaining 90% of images. Then, we combine these pseudo-labels and the ground-truth labels to train a new part model. As shown in Table A below, this model performs about as well as the one trained with segmentation labels for 100% of training images (3rd vs 4th row or 84.9%/39.8% clean/robust accuracy vs 85.6%/39.4%) and performs significantly better than a model trained with no segmentation labels (3rd vs 1st row or 84.9%/39.8% vs 74.7%/37.7%).

Table A: Effects of using only 10% (2K) part segmentation labels (+ 90% or 18K pseudo-labels) to train the part-based model (the third row). The number of training samples is 20K.
| Models                 | # Train Samples | # Seg. Labels          | Clean Acc. | Adv. Acc. |
|------------------------|-----------------|------------------------|-----------:|----------:|
| ResNet-50 (baseline)   | 20K             | None                   |       74.7 |      37.7 |
| Downsampled part model | 20K             | 2K (GT)                |       78.7 |      38.9 |
| **Downsampled part model** | **20K**             | **2K (GT) + 18K (pseudo)** |       **84.9** |      **39.8** |
| Downsampled part model | 20K             | 20K (GT)               |       85.6 |      39.4 |

The difference between this model and the experiments from Section 5.3 and Figure 6 of our paper is that the old models do not use any form of pseudo-labeling. **This suggests that our part-based models benefit from (even simplistic) semi-supervised learning, improving clean and adversarial accuracy by 6% and 1% respectively** (2nd vs 3rd row in Table A). It is possible that applying state-of-the-art methods for semi-supervised learning might yield further improvement. All in all, these experiments offer hope that it may be possible to significantly reduce the labeling cost (e.g., using bounding-box labels, labeling only a small set of images), even on large datasets, while still obtaining good improvements to robustness.

---

> ### Author Response · Authors · 2022-11-19
> **Common response to all reviews (2/2)**
>
> Next, to test scaling, we double the training set size (from 20K to 40K) of PartImageNet by drawing additional samples from ImageNet, with class labels but no additional segmentation labels. Table B compares our part model to the baseline where a model is trained with this extra data but no segmentation labels. It shows that our part model scales well with more training data: **it benefits from extra training data similarly to the normal model and still outperforms it by a large margin (10% clean and 3% adversarial accuracy).** Here, the effective number of part segmentation labels is only 5% of all training samples (2K of 40K).
>
> Table B: Effects of doubling the number of training samples (40K) while keeping the same number of ground-truth segmentation labels (2K).
> | Models                 | # Train Samples | # Seg. Labels          | Clean Acc. | Adv. Acc. |
> |------------------------|-----------------|------------------------|-----------:|----------:|
> | ResNet-50 (baseline)   | 40K             | None                   |       77.7 |      41.9 |
> | **Downsampled part model** | **40K**             | **2K (GT) + 38K (pseudo)** |       **87.1** |      **44.5** |
>
> #### Q: Attacking the segmenter directly does not work well (WbK2, moqQ).
>
> The key difference in our work is that our segmenter is adversarially trained (end-to-end, together with the classifier). As a result, our segmenter is significantly more robust than the undefended segmenters studied in prior work.
>
> We have experimented extensively with directly attacking the segmenter. These attacks work less well than attacking the part model end-to-end. See the responses to Review WbK2 and moqQ for further details.
>
> #### Q: Performance with other backbone models (cNQX, 8aoz).
>
> We train both the baseline and our part models with two additional backbone architectures with a similar size to ResNet-50 (EfficientNet B4 and ResNeXt-50 32x4d). **We find that our part model consistently improves over the baseline across all architectures (5-9% increase in clean and 3-4% in adversarial accuracy).**
>
> Table E: Comparison of other backbone architectures.
> | Backbone Arch.   | Models     | Clean Acc. | Adv. Acc. |
> |------------------|------------|-----------:|----------:|
> | EfficientNet B4  | Baseline   |       83.1 |      37.1 |
> | EfficientNet B4  | Part Model |       88.4 |      41.4 |
> | ResNeXt-50 32x4d | Baseline   |       77.4 |      36.9 |
> | ResNeXt-50 32x4d | Part Model |       86.4 |      39.6 |

---

### Decision · Program_Chairs · 2023-01-20

**Decision:**

Accept: poster

**Justification For Why Not Higher Score:**

The paper proposes an interesting idea but the presentation of the experimental findings is not clear in all aspects.

**Justification For Why Not Lower Score:**

Overall, reviewers and AC agree that the proposed approach is sufficiently interesting, so that the paper can be acceptd to ICLR.

**Metareview: Summary, Strengths And Weaknesses:**

The paper proposes a heuristic approach that leverages part segmentations to increase the adversarial robustness for object classification. The proposed idea to leverage parts of ojects in networks explicitly is interesting and the achieved improvements under adversarial training using PGD or TRADES are very significant.
Yet, the initial presentation of the experimentatl results of  paper was unclear.
During the discussion, many additional results have been reported so that the review scores have increased on average. After the author-reviewer discussion, the paper is still a borderline case, where several aspects are not very clear.
We strongly recommend the authors to discuss the aspect as of  why the results in table 1 improve more strongly for clean data than for adversarial attacks, and to add the baseline results without adversarial training to the table.

**Note From Pc:**

if the above contains the word "oral" or "spotlight" please see: "oral" presentation means -> notable-top-5% and "spotlight" means -> notable-top-25%. As stated in our emails, we are disassociating presentation type from AC recommendations

**Summary Of Ac-Reviewer Meeting:**

During the virtual disucssion amongst reviewers, many aspects could be clarified, e.g. why the results in table 1 improve more strongly for clean data.
We strongly recommend the authors to discuss this aspect and to add the baseline results without adversarial training to the table.